# Integrated exome and RNA sequencing of *TFE3*-translocation renal cell carcinoma

Guangxi Sun [1,11], Junru Chen [1,11], Jiayu Liang[1,11], Xiaoxue Yin[2,11], Mengni Zhang[2], Jin Yao [3], Ning He[4], Cameron M. Armstrong[5], Linmao Zheng[2], Xingming Zhang[1], Sha Zhu[1], Xiaomeng Sun[4], Xiaoxia Yang[4], Wanbin Zhao[4], Banghua Liao[1], Xiuyi Pan[2], Ling Nie[2], Ling Yang[3], Yuntian Chen[3], Jinge Zhao[1], Haoran Zhang[1], Jindong Dai[1], Yali Shen[6], Jiyan Liu[7], Rui Huang[8], Jiandong Liu[1,9], Zhipeng Wang[1], Yuchao Ni[1], Qiang Wei[1], Xiang Li[1], Qiao Zhou[2], Haojie Huang [10], Zhenhua Liu [1,12✉], Pengfei Shen [1,12✉], Ni Chen [2,12✉] & Hao Zeng [1,12✉]

*TFE3*-translocation renal cell carcinoma (*TFE3*-tRCC) is a rare and heterogeneous subtype of kidney cancer with no standard treatment for advanced disease. We describe comprehensive molecular characteristics of 63 untreated primary *TFE3*-tRCCs based on whole-exome and RNA sequencing. *TFE3*-tRCC is highly heterogeneous, both clinicopathologically and genotypically. *ASPSCR1-TFE3* fusion and several somatic copy number alterations, including the loss of 22q, are associated with aggressive features and poor outcomes. Apart from tumors with *MED15-TFE3* fusion, most *TFE3*-tRCCs exhibit low PD-L1 expression and low T-cell infiltration. Unsupervised transcriptomic analysis reveals five molecular clusters with distinct angiogenesis, stroma, proliferation and KRAS down signatures, which show association with fusion patterns and prognosis. In line with the aggressive nature, the high angiogenesis/stroma/proliferation cluster exclusively consists of tumors with *ASPSCR1-TFE3* fusion. Here, we describe the genomic and transcriptomic features of *TFE3*-tRCC and provide insights into precision medicine for this disease.

[1] Department of Urology, Institute of Urology, West China Hospital, Sichuan University, Chengdu, China. [2] Department of Pathology, West China Hospital, Sichuan University, Chengdu, China. [3] Department of Radiology, West China Hospital, Sichuan University, Chengdu, China. [4] GloriousMed Clinical Laboratory (Shanghai) Co., Ltd, Shanghai, China. [5] Department of Urology and Comprehensive Cancer Center, University of California Davis, Sacramento, CA, USA. [6] Department of Oncology, West China Hospital, Sichuan University, Chengdu, China. [7] Department of Biotherapy, West China Hospital, Sichuan University, Chengdu, China. [8] Department of Nuclear medicine, West China Hospital, Sichuan University, Chengdu, China. [9] Department of Urology, The First Affiliated Hospital, School of Medicine, Zhejiang University, Hangzhou, China. [10] Departments of Biochemistry and Molecular Biology and Urology, Mayo Clinic College of Medicine and Science, Rochester, MN, USA. [11] These authors contributed equally: Guangxi Sun, Junru Chen, Jiayu Liang, Xiaoxue Yin. [12] These authors jointly supervised this work: Zhenhua Liu, Pengfei Shen, Ni Chen, Hao Zeng. ✉email: zhliu@scu.edu.cn; cdhx510@163.com; chenni1@163.com; kucaizeng@163.com

TFE3-translocation renal cell carcinoma (TFE3-tRCC) is a rare subtype of kidney cancer, characterized by Xp11.2 translocations resulting in TFE3 fusion with various partner genes. To date, more than 20 partner genes have been identified in fusions with TFE3, including SFPQ, ASPSCR1, NONO, PRCC, RBM10, MED15, etc. Due to the variety of partners and fusion structures, the function of chimeric TFE3 fusion proteins is diverse, which may contribute to the high degree of heterogeneity of TFE3-tRCC, both morphologically and clinically[1–5]. Although only 1–4% RCCs in adults may have TFE3 translocations[6,7], TFE3-tRCC constitutes 15% of RCCs in patients <45 years of age and 20–50% of pediatric RCCs[8,9]. Unfortunately, owing to the limited understanding of the underlying mechanisms[10], the optimal therapy for TFE3-tRCC remains to be determined, prompting an urgent clinical need to molecularly characterize TFE3-tRCC.

Before this study, our knowledge concerning the molecular features of TFE3-tRCC was limited as previous analyses contained relatively small cohorts and were mainly confined to genomic alterations[11–14]. Although TFE3-tRCC was found to have few recurrent mutations, prior reports revealed the association between the somatic copy number alterations (SCNA) and survival outcomes[11–13]. However, these genomic alterations found in tumors were not enough to explain the high heterogeneity of TFE3-tRCC. The comprehensive genomic, especially the transcriptomic, characteristics of TFE3-tRCC are still uncovered.

In this study, we apply whole-exome sequencing (WES) on 53 TFE3-tRCCs and RNA sequencing (RNA-seq) on 63 TFE3-tRCCs to reveal their genomic and transcriptomic characteristics and discover molecular mechanisms potentially involved in tumor progression. Our analyses reveal the prognostic value of ASPSCR1-TFE3 fusion and SCNAs for TFE3-tRCC, identify five molecular subsets with distinct transcriptional signatures.

## Results

**Identification and clinicopathologic features of TFE3-tRCC.** From 2009 to 2019, a total of 68 patients with TFE3-tRCC were identified from 4581 RCC cases (Supplementary Figure 1).

**Table 1 Baseline clinicopathologic characteristics of patients with TFE3-tRCC.**

| Clinicopathologic characteristics | Total (n = 68) |
|---|---|
| Age, median (range) | 32.5 (5–70) |
| **Gender, n (%)** | |
| Male | 26 (39.7%) |
| Female | 42 (60.3%) |
| **Tumor size, median (cm, range)** | 4.7 (1.4–19.6) |
| **T stage, n (%)** | |
| ≤T2 | 59 (86.8%) |
| ≥T3 | 9 (13.2%) |
| **N stage, n (%)** | |
| N0 | 52 (76.5%) |
| N1 | 16 (23.5%) |
| **M stage, n (%)** | |
| M0 | 61 (89.7%) |
| M1 | 7 (10.3%) |
| **ISUP grade, n (%)** | |
| ≤ 2 | 31 (45.6%) |
| ≥ 3 | 37 (54.4%) |
| **Nephrectomy, n (%)** | |
| Nephron-sparing surgery | 27 (39.7%) |
| Radical nephrectomy | 41 (60.3%) |

TFE3-tRCC TFE3-translocation renal cell carcinoma, ISUP The International Society of Urological Pathology.

The baseline clinicopathologic characteristics of the cases with TFE3-tRCC are summarized in Table 1 and Supplementary Data 1. The median age at diagnosis was 32.5 years (range: 5–70 years) and nine (13.2%) patients were younger than 18 years old. The male: female ratio was 2:3. The median tumor size was 4.7 cm (range: 1.4–19.6 cm). At initial diagnosis, 16 (23.5%) and seven (10.3%) patients presented with regional lymph node metastasis and distant metastasis, respectively. Morphologically, TFE3-tRCCs presented with diverse architectural and cytologic features, including papillary, tubular, acinar, and cystic patterns and 37 (54.4%) tumors had an international society of urological pathology (ISUP) grade ≥3. For primary kidney tumors, 27 (39.7%) and 41 (60.3%) patients underwent nephron-sparing surgery and radical nephrectomy, respectively. Ten (14.7%) patients died at the end of follow-up (median 43.8 months, 95% CI: 31.5–56.1).

**Identification of TFE3-fusion partners and structures.** RNA-seq was performed on 63 TFE3-tRCC tumors and 14 adjacent normal kidney tissues. Gene fusions were detected in 90.5% (57/63) cases, of which 94.7% (54/57) cases showed relatively more common gene fusions, including SFPQ-TFE3 (n = 15), ASPSCR1-TFE3 (n = 13), NONO-TFE3 (n = 8), MED15-TFE3 (n = 8), PRCC-TFE3 (n = 6), and RBM10-TFE3 fusions (n = 4) (Fig. 1A). Three (5.3%, 3/57) cases were identified with rare TFE3-associated gene fusions, including FUBP1-TFE3, SETD1B-TFE3, and ZC3H4-TFE3, of which FUBP1-TFE3 and SETD1B-TFE3 fusion showed previously unreported fusion structures. FUBP1-TFE3 fusion resulted in a chimeric transcript composed of exons 1–15 of FUBP1 and exons 3–10 of TFE3, and SETD1B-TFE3 fusion was composed of exons 1–4 of SETD1B and exons 4–10 of TFE3. All three rare gene fusions were confirmed by RT-PCR and Sanger sequencing (Supplementary Figure 2). Patients with SETD1B-TFE3 and ZC3H4-TFE3 fusion developed metastasis by the end of follow-up.

The structures of the TFE3 fusion isoforms identified in our cohort are summarized in Fig. 1B and C. All fusion genes preserved the open reading frame between the 5′ terminal of partner genes and the 3′ terminal of TFE3. According to the retained exons and functional domains of the TFE3, six types of isoforms were found, including retained fragment of TFE3 2–10 exons (5.3%, 3/57), 3–10 exons (3.5%, 2/57), 4–10 exons (15.8%, 9/57), 5–10 exons (22.8%, 13/57), 6–10 exons (50.9%, 29/57), and 7–10 exons (1.7%, 1/57). All fusions retained exons 7–10 of the TFE3 gene, containing the helix–loop-helix (bHLH) and leucine zipper (LZ) domains, but only a subset (47.4%, 27/57) of fusion isoforms contained the transcription activation (AD) domain.

The fusion isoforms of different TFE3-tRCC subtypes were highly heterogeneous (Fig. 1C and Supplementary Data 2). Among which, SFPQ-TFE3 fusion isoforms had a maximum of six fusion structures, in contrast, ASPSCR1-TFE3 and RBM10-TFE3 isoforms had relatively fixed fusion structures. Next, we analyzed the functional domains of the retained exons of the fusion partner genes (Fig. 1C). We found 42% of fusion partner genes retained all functional domains. Interestingly, fusion partners that play regulatory roles in mRNA processing and/or mRNA splicing, including NONO, SFPQ, and RBM10, retained all RNA-recognition motifs (RRM).

**TFE3-fusion partners impact tumor phenotype and survival outcomes.** We subsequently explored the association of clinicopathologic features with different TFE3 fusion subtypes. Morphologically, we found tumors harboring ASPSCR1-TFE3 fusion showed a predominated papillary pattern, and tumors with MED15-TFE3 fusion showed a dominant multicystic pattern

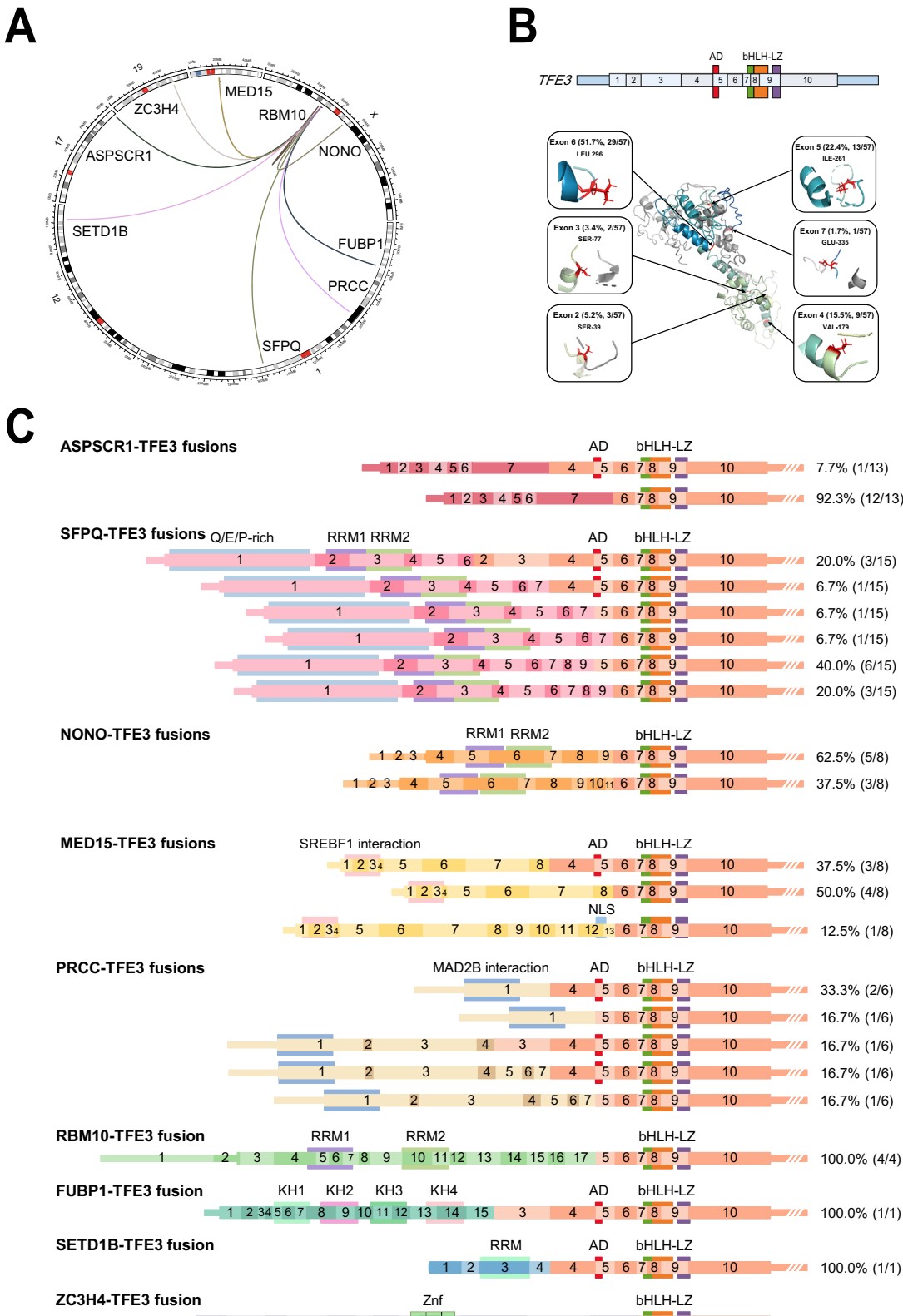

**Fig. 1 TFE3 fusion isoforms and structures. A** The circle represents *TFE3* gene fusions in our *TFE3*-tRCC cohort. **B** Exons and function domains for *TFE3* gene (top). Protein domains are color-coded and depicted. Mapping of breakpoints onto the three-dimensional structure of TFE3 protein (bottom). **C** Exons and functional domains of the *TFE3* and fusion partner genes detected in our *TFE3*-tRCC cohort. *TFE3*-tRCC *TFE3*-translocation renal cell carcinoma, AD strong transcription activation domain, bHLH basic helix–loop–helix domain, LZ leucine zipper domain, RRM RNA-recognition motif, SREBF1 Sterol Regulatory Element Binding Transcription Factor 1, MAD2L2 mitotic spindle assembly checkpoint protein MAD2B, KH K homology domain, Znf zinc-finger domains.

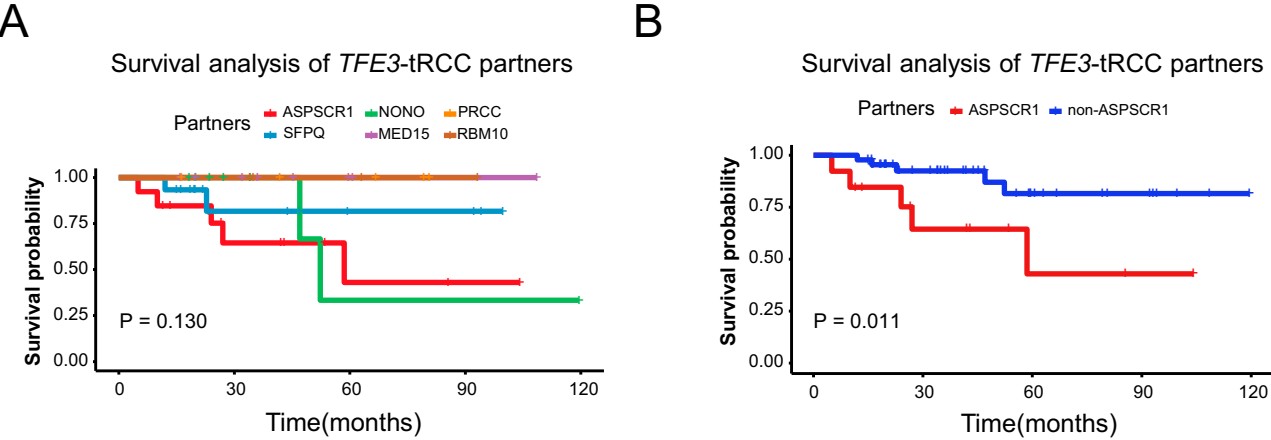

**Fig. 2 Survival analysis of *TFE3*-tRCC partners. A** Kaplan–Meier curves show the OS for patients with different *TFE3*-fusion subtypes. **A** Kaplan–Meier curves show the OS for patients with *ASPSCR1-TFE3* fusion and non- *ASPSCR1-TFE3* fusion. *P*-value was determined by two-sided log-rank test (**A**, **B**). *TFE3*-tRCC *TFE3*-translocation renal cell carcinoma, OS overall survival.

(Supplementary Figure 3A). Radiologically, cases with *ASPSCR1-TFE3* fusion presented with features typical of hypervascularity and calcification on CT scan, while *MED15-TFE3* fusion tRCC often presented as cystic masses (Supplementary Figure 3B).

Of note, compared with other *TFE3*-tRCC subtypes, tumors with *ASPSCR1-TFE3* fusion were highly aggressive, characterized by higher ISUP nuclear grade (ISUP $\geq$ 3, 11/13 vs. 19/44, $P = 0.017$) and more frequent lymph node metastasis (6/13 vs. 7/44, $P = 0.057$, Supplementary Table 1). At the end of follow-up, a total of 20 patients had presented with distal metastasis, of which over one-third were *ASPSCR1-TFE3* fusion tRCCs (7/20, 35%). Survival analysis showed that those with *ASPSCR1-TFE3* fusion were associated with poor overall survival (OS) compared with other subtypes (median OS: 58.5 months vs. not reached, $P = 0.011$, Fig. 2A and B). In multivariate analysis, after adjusting for clinicopathologic features, the *ASPSCR1-TFE3* fusion was still independently associated with poor outcome ($P = 0.010$, Supplementary Table 2 and Supplementary Figure 4).

**Somatic mutational landscape of *TFE3*-tRCC.** WES was performed on 53 *TFE3*-tRCCs and matched germline samples. We identified a total of 1591 somatic mutations, including 1486 nonsynonymous single-nucleotide variants (SNVs) and 105 insertions/deletions (Indels), with a median of 14 (range: 0–238) mutations per tumor (Fig. 3A). The tumor mutation burden (median 0.28, range: 0–4.76) in our *TFE3*-tRCC cohort was lower than that of KIRC and KIRP in TCGA cohort (Supplementary Figure 5). We extracted three prominent mutational signatures using a non-negative matrix factorization (NMF) algorithm (Supplementary Figure 6). Signature B shows the largest contribution, which is found to be similar to SBS40, a signature correlated with age in multiple types of cancer[15]. Signature A is similar to both the SBS87 (thiopurine exposure) and SBS1 (age-related 5-methylcytosine deamination). Signature C, which highly corresponds to SBS22 (cosine similarity = 0.91), is characterized by T > A transversions at CT [A/G] and has been associated with aristolochic acid exposure. We observed Signature C in 28.8% of patients in our cohort, indicating a potential role of aristolochic acid exposure in the development of Chinese *TFE3*-tRCC. The frequently mutated genes (frequency of more than four samples) included *DST*, *DNAH8*, and *HMHA1*, whereas the mutated loci at each gene were not recurrent (Fig. 3B and Supplementary Data 3). In addition, six tumor suppressor genes previously implicated in cancer (*BTK*, *CHD1*, *FN1*, *NFATC2*, *NOTCH1*, and *NRP1*) were found in at least two samples (Fig. 3B). Of these, 75%

(9/12) were clonal mutations (Supplementary Data 3). In line with previous studies, the mutational spectrum of *TFE3*-tRCC was quite heterogeneous. Survival analysis indicated no relation between frequent SNVs and patient survival (Supplementary Table 3).

The most frequently observed individual arm-level events included gain of 17q (12/53, 23%) and 19p (11/53, 21%), and loss of 19p (16/53, 30%), 14q (14/53, 26%), and 1p (11/53, 21%). The most frequent focal events were gain of 19p13.2 (17/53, 32%), 1q44 (15/53, 28%), and 8q24.3 (13/53, 25%), and loss of 19p12 (15/53, 26%), 14q21.2 (13/53, 25%, Fig. 3C and Supplementary Data 4). Previous studies reported that certain copy number events (eg. 9p loss and 17q gain) were correlated with patient outcomes, therefore, we evaluated the association of SCNA with clinicopathologic features and prognosis in our *TFE3*-tRCC cohort (Supplementary Table 4). We found that tumors with 22q loss were correlated with *ASPSCR1-TFE3* fusion (4/4 vs. 2/38, $P = 0.005$), higher ISUP nuclear grade (ISUP $\geq$ 3, 7/19 vs. 0/27, $P = 0.004$) and more frequent lymph node metastasis (5/7 vs. 2/39, $P = 0.004$). Cases with 9p loss were associated with increased lymph node metastasis (4/8 vs. 2/39, $P = 0.019$). Survival analysis indicated that loss of chromosome arms 1p, 2p, 6q, 8p, 9p, and 22q were predictors for poor OS (Supplementary Figure 7 and Fig. 3D). Moreover, we identified that tumors with higher SCNA burden significantly correlated with worse survival outcomes (median OS: 59.46 months vs. 111.28 months, $P = 0.006$). After adjustment for clinicopathologic features, 22q loss was identified as an independent predictor for poor OS ($P = 0.004$, Supplementary Table 2 and Supplementary Figure 4).

**Transcriptional landscape and immune microenvironment features of *TFE3*-tRCC.** Analysis of differentially expressed genes (DEG) identified a total of 3124 over-expressed and 2143 under-expressed genes in *TFE3*-tRCCs compared to adjacent normal tissues (Fig. 4A and Supplementary Data 5). Among them, *GPNMB*, *HIF1A*, *MET*, and *BIRC7*, which were previously reported to have higher expression in tRCC[16,17], were also upregulated in our *TFE3*-tRCC cohort (Supplementary Figure 8A). Kyoto Encyclopedia of Genes and Genomes (KEGG) pathway analysis identified that lysosomal, autophagy, and innate immunity pathways were significantly upregulated in *TFE3*-tRCC (Fig. 4B). Gene set enrichment analysis (GSEA) demonstrated enrichment of transcriptional pathways involved in proliferation (E2F targets and G2M checkpoint), PI3K/ATK/mTOR, and p53 signaling in *TFE3*-tRCC (Fig. 4C).

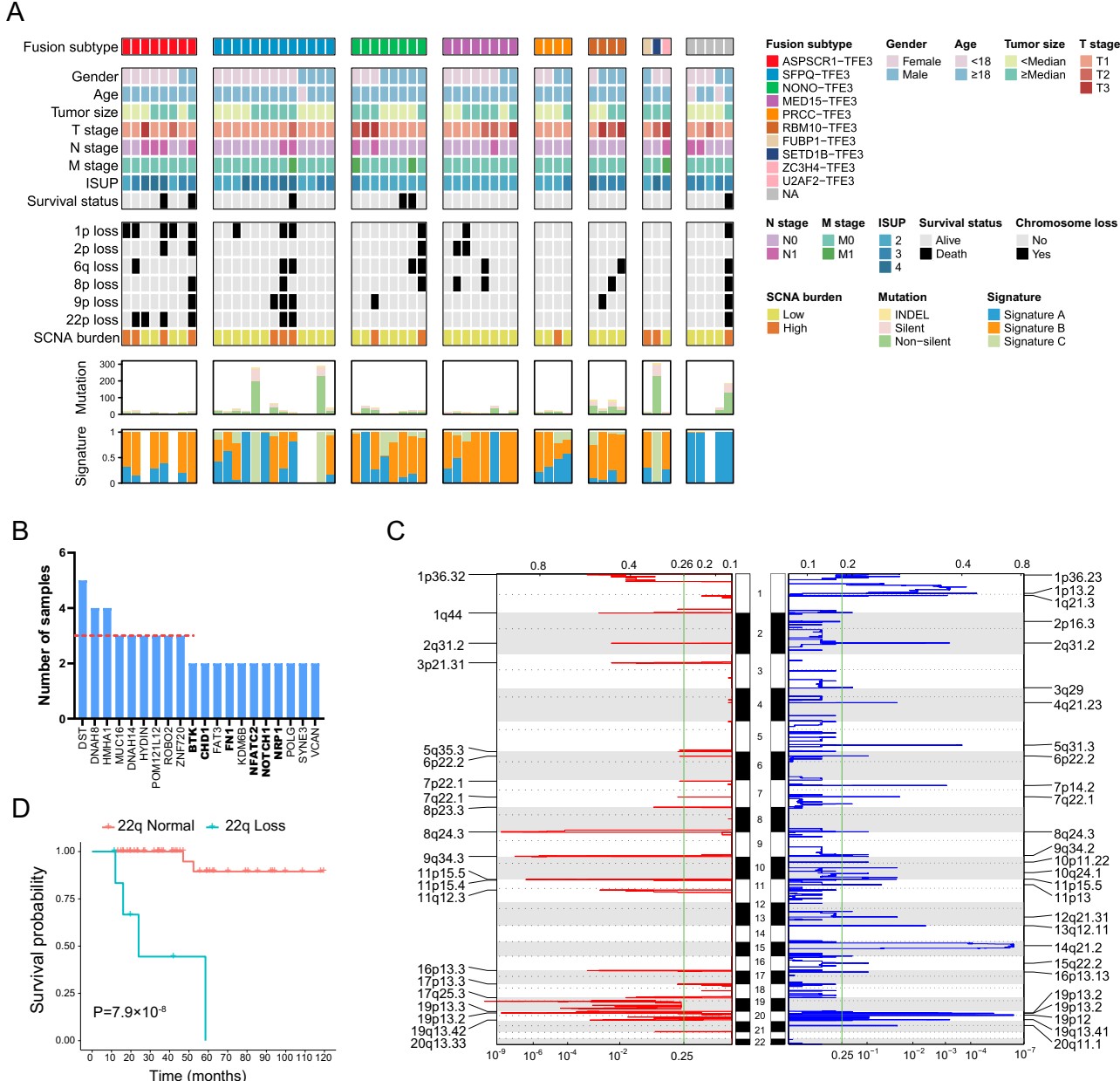

**Fig. 3 The mutational landscape of TFE3-tRCC. A** Clinical features and molecular data for 53 tumors (rows) are displayed as heatmaps. Each column represents an individual tumor. Patients were separated into eight groups: those with *ASPSCR1-TFE3* (n = 8), with *SFPQ-TFE3* (n = 13), with *NONO-TFE3* (n = 8), with *MED15-TFE3* (n = 8), with *PRCC-TFE3* (n = 4), with *RBM10-TFE3* fusions (n = 4), with rare *TFE3*-associated gene fusions (n = 3, including *FUBP1-TFE3*, *SETD1B-TFE3*, and *ZC3H4-TFE3*) and unknow partners (n = 5). The top panel shows fusion partners. The middle panel shows clinical features, including gender, age, tumor size, TNM stage, ISUP, survival status, SCNA, and SCNA burden. Each subsequent panel displays a specific molecular profile: number of mutations and the fraction of each COSMIC signature in the genome. **B** Frequently mutated genes in the *TFE3*-tRCC cohort. The red dashed line denotes three mutated patients. Tumor suppresser genes are labeled with bold font. **C** Focal amplification and deletion determined from GISTIC 2.0 analysis. The green line indicates the cut-off for significance (q = 0.25). P values significantly peaks were identified at FDR q value <0.25. The top and bottom numerical values refer to G-scores and q values, respectively. **D** Kaplan–Meier curves show the OS between the patients with 22p loss and 22p normal. P value was determined by two-sided log-rank test. *TFE3*-tRCC *TFE3*-translocation renal cell carcinoma, ISUP International Society of Urological Pathology, SCNA somatic copy number alteration, OS overall survival.

Next, we explored the tumor immune microenvironment (TIME) in *TFE3*-tRCC. We found a lower immune infiltration score (IIS) and T-cell infiltration score (TIS) in *TFE3*-tRCC in contrast to KIRC from the TCGA cohort (Fig. 4D and Supplementary Figure 8B). Using a refined RCC immune cell gene-specific signatures[18], we found that compared with TCGA RCC subtypes, the T helper 2 cell (Th2) signature was increased in *TFE3*-tRCC, while the activated dendritic cell (aDC) and plasmacytoid dendritic cell (pDC) signatures had decreased expression (Fig. 4E and Supplementary Figure 8B). Moreover, the natural killer cell (NK) signature was increased in most *TFE3*-tRCC compared with KIRP and KICH. Furthermore, lower levels of CD8$^+$ T-cell, T-cell, and macrophage signatures were identified in *TFE3*-tRCC relative to KIRC (Fig. 4E). Consistently, immunohistochemistry (IHC) confirmed that two-thirds (74.6%, 47/63) of tumors had low CD8$^+$ T-cells infiltrations, while only

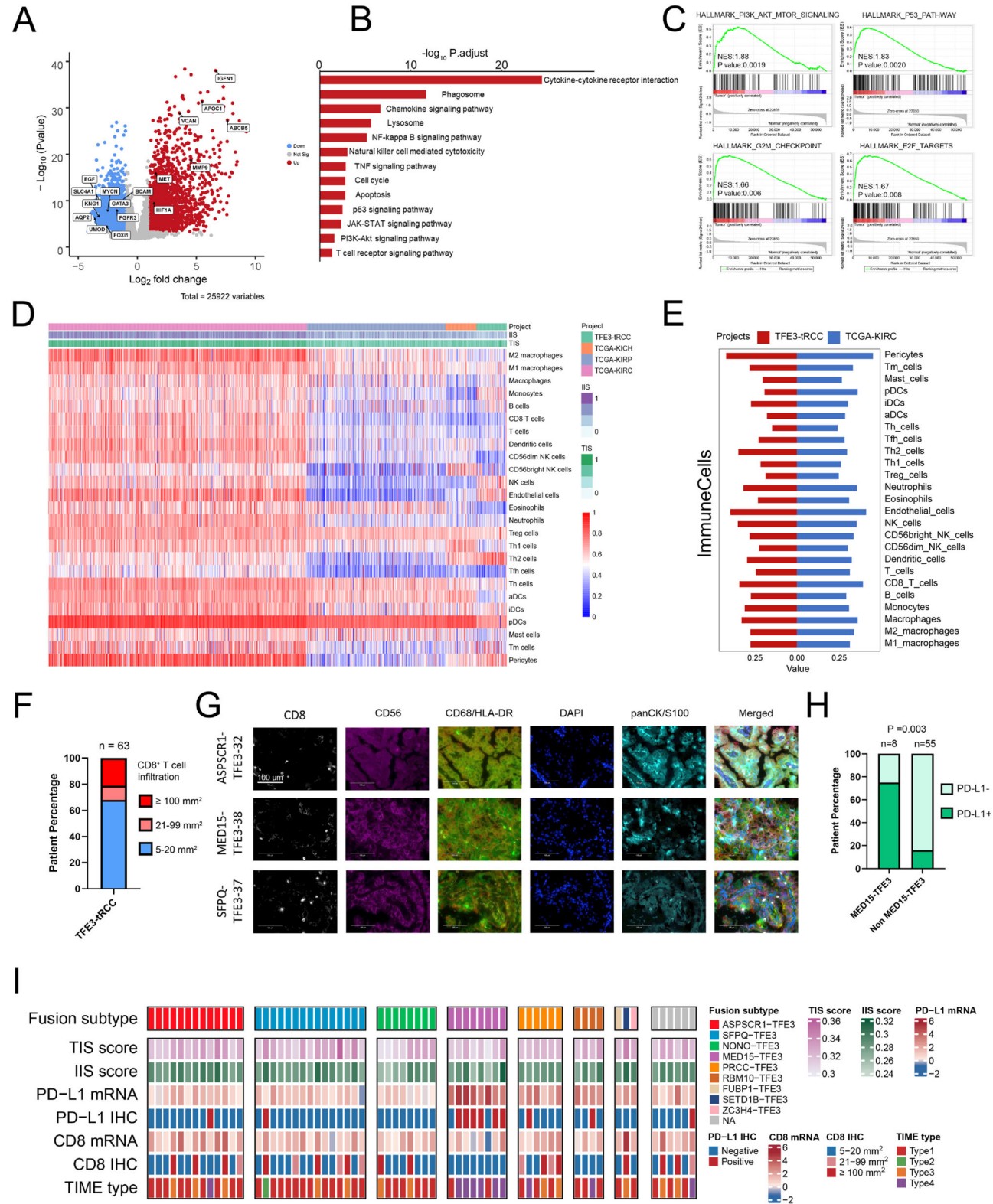

twenty percent (20.6%, 13/63) of tumors showed massive CD8+ T cells infiltrations (Fig. 4F and Supplementary Figure 9). In tumors with a low CD8+ T-cells infiltrations, multiple immunofluorescence further revealed a low infiltration of NK (CD56+) cells and macrophages (CD68+/HLA-DR+) in the tumor stroma (Fig. 4G). In line with a previous analysis[13], the expression of PD-L1 mRNA was higher in TFE3-tRCC than KIRC and KIRP in our analysis (Supplementary Figure 8 C). However,

different from the RNA-seq results, there was a relatively low PD-L1 positivity rate (17.5%, 11/63) in our TFE3-tRCC cohort as assessed via PD-L1 IHC staining. Interestingly, we observed higher expression of PD-L1 mRNA and protein levels in tumors with MED15-TFE3 fusion (Fig. 4H and Supplementary Figure 8D). Apart from MED15-TFE3 fusion tRCC, nearly two-thirds (63.5%, 40/63) of TFE3-tRCCs in our cohort exhibited type 1 TIME[19], characterized by a low CD8+ T-cells infiltration and low

**Fig. 4 Transcriptional features and immunogenic phenotype in TFE3-tRCC. A** A volcano plot representing differentially expressed genes between the tumors and paired adjacent normal tissues. Significantly upregulated genes are colored in red, whereas significantly downregulated genes are colored in blue. The q values (-log10 p) were calculated using paired two-sided moderated Student's t test. **B** Kyoto Encyclopedia of Genes and Genomes enrichment analysis of differentially expressed genes. Enrichment q values (-log10 p) are calculated by hypergeometric test. **C** GSEA enrichment scores of PI3K-AKT-mTOR, P53, G2M Checkpoint, and E2F targets pathways in TFE3-tRCC versus normal tissues. **D** Unsupervised clustering of samples from the TCGA RCC and our TFE3-tRCC cohorts (n = 63) using ssGSEA scores from 25 immune cell types, IIS and TIS. KICH (n = 65), KIRC (n = 539), KIRP (n = 289). **E** Comparing the expression of 25 immune cell types, IIS and TIS between TCGA-KIRC (n = 539) and our TFE3-tRCC cohorts (n = 63). **(F)** Bar chart depicts the prevalence of CD8 expression by immunohistochemistry. **G** Representative immunofluorescence demonstrating the presence of overall CD8 + T-cell, tumor-associated macrophage (CD68 and HLA-DR), and NK cell (CD56) infiltration in three selected samples (TFE3-32, TFE3-38, and TFE3-37) in our cohort. Ten random high-power fields of tumor parenchyma were checked for CD8, CD68, HLA-DR, and CD56 positive expression. Magnification × 400. Scale bar = 100 μm. **H** Bar chart depicts the prevalence of PD-L1 expression by immunohistochemistry. P-values were determined by Pearson's chi-square test. **I)** Heatmap shows the immune features among TFE3-tRCCs (n = 63). Each column represents an individual tumor. Patients were separated into eight groups: those with ASPSCR1-TFE3 (n = 13), with SFPQ-TFE3 (n = 15), with NONO-TFE3 (n = 8), with MED15-TFE3 (n = 8), with PRCC-TFE3 (n = 6), with RBM10-TFE3 fusions (n = 4), with rare TFE3-associated gene fusions (n = 3, including FUBP1-TFE3, SETD1B-TFE3, and ZC3H4-TFE3), and unknow partners (n = 6). The top panel shows fusion partners and isoforms. The bottom panel shows immune features, including TIS, IIS, PD-L, CD8, and TIME type. TFE3-tRCC TFE3-translocation renal cell carcinoma, TCGA The Cancer Genome Atlas. KICH chromophobe renal cell carcinoma, KIRC renal clear cell carcinoma, KIRP renal papillary cell carcinoma, IIS immune infiltration score, TIS T-cell infiltration score, IHC immunohistochemistry, TIME tumor immune microenvironment.

PD-L1 expression in tumors (Fig. 4I). Taken together, our data demonstrated an immunologically ignorant microenvironment in TFE3-tRCC.

**Identification and characterization of five molecular subtypes of TFE3-tRCC.** To move forward on understanding the biology of TFE3-tRCC, we performed unsupervised clustering analysis using NMF to identify and refine transcription-based subgroups of patients with common TFE3-tRCC subtypes (n = 54). Consequently, five distinct clusters were identified based on the 1500 most variable genes in our TFE3-tRCC cohort (Fig. 5A). To further reveal the transcriptional features driving these clusters, we performed quantitative set analyses for gene expression (QuSAGE)[20] to compare each cluster with all others utilizing Hallmark gene sets (Fig. 5B and Supplementary Data 6). Combining this with the results of DEG analysis, which also compared clusters individually to others, we summarized and refined four signatures of representative genes involving stroma, angiogenesis, proliferation and KRAS down (Fig. 5C). Tumors in cluster 5 were characterized as highly angiogenic and proliferative, which showed high expression of angiogenesis-related genes (e.g., VEGFA, POSTN and HIF1A) and had enrichment of cell-cycle transcriptional programs (E2F targets, G2M checkpoint, and p53). Compared with tumors in the other clusters, tumors in cluster 5 had higher angiogenesis scores (Fig. 5D). IHC further confirmed that these tumors had the strongest nuclear HIF1A staining and increased vessel density (Fig. 5E). Moreover, tumors in cluster 5 also exhibited higher expression of stroma signature, exemplified by significantly higher epithelial-mesenchymal transition (EMT) score, and a high degree of collagens and activated stroma-related genes (e.g., FAP, CDH2, and CDH6; Fig. 5C and D). Cluster 1 was characterized by moderate enrichment of EMT, apical junction, TGF-β, WNT catenin, and hypoxia signaling, which have been associated with stroma and angiogenesis signatures (Fig. 5B). Tumors in cluster 3 and 4 were featured by low expression of angiogenesis modules (Fig. 5B–D). In contrast to cluster 3, cluster 4 showed moderate enrichment of E2F signaling. Cluster 2 differentiated from the other clusters by enrichment of genes repressed by KRAS down signature (Fig. 5B and C).

**TFE3-tRCC molecular subtypes associate with patient prognosis and efficacy to systematic treatment.** Remarkably, we demonstrated that this molecular classification was associated with fusion subtypes and patient outcomes (Supplementary Figure 10). All tumors with ASPSCR1-TFE3 fusion were classified into the high angiogenesis/stroma/proliferation cluster (cluster 5), which exhibited worse survival. Cluster 1 included a half of MED15-TFE3 (62.5%, 5/8) and SFPQ-TFE3 (46.7%, 7/15) fusion tRCCs and cluster 3 was mainly comprised of NONO-TFE3 (75%, 6/8) and SFPQ-TFE3 (26.7%, 4/15) fusion tRCCs. Both the clusters showed moderate EMT score and intermediate survival (Fig. 5D). Although cluster 2 and cluster 4 consisted of mixed fusion subtypes, tumors in these clusters showed decreased EMT score, and demonstrated more favorable prognosis.

A total of eight patients received systematic therapy in our TFE3-tRCC cohort. Of note, the median progression-free survival (PFS) for patients receiving first-line tyrosine kinase inhibitors (TKIs) treatment was only 3 months (Fig. 6A), confirming that the low angiogenesis signature interfered with the efficacy of TKIs monotherapy for patients with TFE3-tRCC. Interestingly, we observed two patients in cluster 5 had sustained control of disease to TKIs plus immune checkpoint inhibitors, with one (TFE3-68) showing partial response (tumor shrinkage 63.1%) after 5 months of first-line axitinib plus pembrolizumab and the other patient (TFE3-65) demonstrating stable disease for 12 months after third-line axitinib plus pembrolizumab (Fig. 6B). These initial clinical results indicated that tumors in cluster 5 might have a higher likelihood of response to antiangiogenic plus immunotherapy.

## Discussion

This study depicted comprehensive molecular analyses of 63 untreated primary TFE3-tRCCs and investigated the molecular characteristics of this rare but highly heterogeneous RCC entity. Our findings provided important new insights into key genomic and biologic features underlying TFE3-tRCC development and progression, confirmed the promotion role of copy number alterations in tumor progression, revealed the prognostic value of TFE3 fusion patterns, and described five distinct molecular clusters associated with different transcriptional signatures.

We identified a low degree of somatic mutation rate but a widespread existence of SCNAs in TFE3-tRCC. Copy number alterations in TFE3-tRCC have been previously explored but with inconsistent results. In a previous study of 16 tRCCs, 17q gain (44%) was identified as the most common SCNA and found to be a predictor for poor prognosis[11]. More recently, a genomic analysis of 22 tRCCs reported loss of 9p (41%) and gain of 17q (36%) were the most frequent SCNAs, but no single SCNA event was associated with worse prognosis[13]. Such inconsistent results may

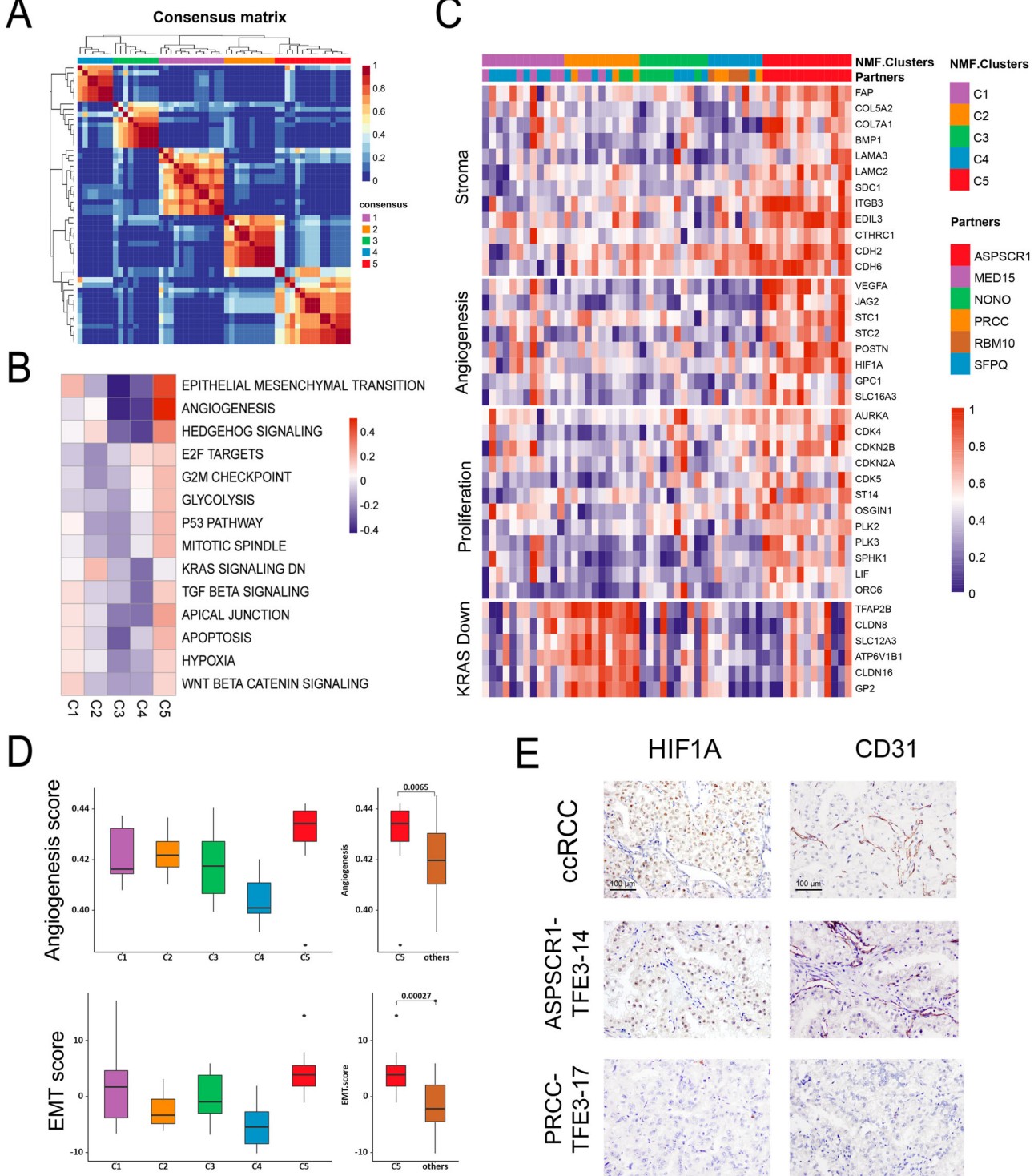

**Fig. 5 Transcriptional stratification identifies TFE3-tRCC tumor subsets with distinct biologic features. A** Consensus matrix depicting clusters ($k = 5$) identified by NMF clustering of tumors with definite fusion partners ($n = 54$). **B** Heatmap representing MSigDb hallmark gene set QuSAGE enrichment scores for each NMF tumor cluster compared with all other tumors. White cells represent non-significant enrichment after FDR correction. **C** Heatmap of genes comprised in transcriptional signatures. Samples are grouped by NMF cluster. **D** Comparative angiogenesis score (top) and EMT score (bottom) in samples between the different NMF tumor clusters (C1–C5). C1, $n = 12$; C2, $n = 11$; C3, $n = 10$; C4, $n = 8$; C5, $n = 13$; others $= C1 + C2 + C3 + C4$, $n = 41$. Box plots show median levels (middle line), 25th and 75th percentile (box), 1.5 times the interquartile range (whiskers) as well as outliers (single points). $P$ values were determined by the two-sides Mann–Whitney $U$ test. **E** Representative IHC demonstrating HIF1A and CD31 expression in two selected samples (TFE3-14 and TFE3-17) in our cohort and one positive control (ccRCC with *VHL* mutant). Five random high-power fields were checked for HIFA and CD31 positive expression. Magnification × 200. Scale bar $= 100 \, \mu m$. *TFE3*-tRCC *TFE3*-translocation renal cell carcinoma, NMF non-negative matrix factorization, EMT epithelial-mesenchymal transition, IHC immunohistochemistry.

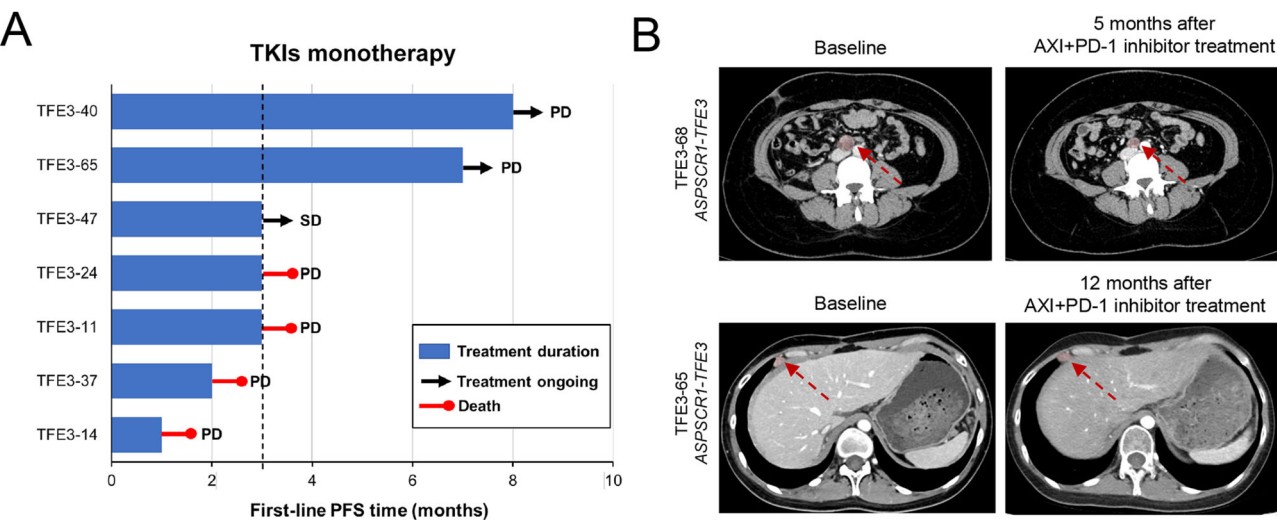

**Fig. 6 Responses to systemic treatment and potential therapeutic targets for patients with TFE3-tRCC. A** Swimmer plot depicts the PFS of individual patients receiving first-line TKIs treatments. Vertical line indicates PFS at 3 months. **B** Baseline imaging in two patients (TFE3-68 and TFE3-65) before initiation of systematic treatment and after they received the combination of pembrolizumab plus axitinib treatment. *TFE3*-tRCC *TFE3*-translocation renal cell carcinoma, TKIs tyrosine kinase inhibitor, PFS progression-free survival. Arrow indicates tumor lesion.

result from the relatively small sample sizes and heterogeneity of *TFE3*-tRCC, since both studies only included a few *TFE3*-tRCC subtypes and were confounded with *TFEB*-tRCC samples. In our cohort, gain of 17q and loss of 9p were found in 23 and 11% of *TFE3*-tRCC, respectively. Loss of chromosome arm 9p but not gain of arm 17q was correlated with poor survival. In addition, loss of 1p, 2p, 6q, 8p, 22q, and increased SCNA burden were also predictors for poor prognosis. More importantly, we demonstrated that 22q loss was significantly associated with aggressive clinical features and an independent predictor of worse outcomes for patients with *TFE3*-tRCC. Increased loss of chromosome 22q that encodes *NF2*, *CHEK2*, and *SMARCB1*[21] were observed in type 2 PRCC, which may implicate in carcinogenesis and tumor progression. A recent study suggested that the presence of copy number variations might be an earlier event in *TFE3*-tRCC than somatic mutations[13]. Therefore, our results again underscore the importance of SCNA in the aggressive characteristics of *TFE3*-tRCC and the prognostic value of copy number aberrations.

Our results demonstrated an immune ignorant TIME in the majority of *TFE3*-tRCCs, characterized by low PD-L1 expression and low CD8[+] T-cell infiltration in tumor stroma. Except for tumors with *MED15-TFE3* fusion, a low PD-L1 positivity rate was detected in our *TFE3*-tRCCs. Therefore, it is possible that the fusion subtype may impact the different PD-L1 expression in *TFE3*-tRCC. Given the low immunogenicity of *TFE3*-tRCC (mostly belonging to TIME type 1), single-agent PD-1/PD-L1 blockade could probably be inadequate to improve outcomes in those with advanced disease, which is supported by several clinical investigations[22–24]. Therefore, combination therapeutic regimens, including radiotherapy, targeted therapy, or chemotherapy plus immunotherapy, could be more effective to mediate immunogenic cell death though inducing neoantigen liberation and lymphocytes recruitment into the tumor microenvironment[25].

*TFE3*-tRCC represents a heterogeneous disease with a wide spectrum of morphologies and varied clinical phenotypes. Several studies suggested that patients with different fusion subtypes exhibited differential prognosis[2]. In this study, we explored the association between fusion subtypes and patient outcomes. We observed that tumors with *ASPSCR1-TFE3* fusion had much worse survival compared to those with other fusions. On the contrary, patients with other common *TFE3* fusions, including

*MED15-TFE3* and *PRCC-TFE3* showed a better prognosis. Our findings highlighted the importance of the detection of *TFE3* fusion partners for predicting prognosis. Also, the detailed mechanisms of *TFE3* fusion partners in regulating the chimeric protein are intriguing and warrant further investigation.

Our unsupervised transcriptomic analysis identified five molecular clusters. Tumors with *ASPSCR1-TFE3* fusion were exclusively classified into cluster 5, exhibiting high expression of angiogenesis/stroma/proliferation gene signatures. The distinct genomic features of *ASPSCR1-TFE3* fusion tRCC identified in the current study suggested a molecular basis for the aggressive nature of this *TFE3*-tRCC subtype. A preclinical study suggested that *ASPSCR1-TFE3* fusion induced accumulation of lactate in tumor microenvironment, leading to tumor hypoxia and angiogenesis in alveolar soft part sarcoma[26]. In agreement with this finding, we found that tumors with *ASPSCR1-TFE3* fusion exhibited strong nuclear HIF1A staining and increased vessel density. Thus, patients with *ASPSCR1-TFE3* fusion tRCC may benefit from antiangiogenic based treatment. It is important to note that the high stroma and proliferation signatures may also interfere with the efficacy of antiangiogenic monotherapy. Interestingly, we observed a favorable response to anti-PD1 plus TKI combination therapy in two *ASPSCR1-TFE3* fusion tRCC patients in this study. Therefore, our data may support clinical investigation of antiangiogenic therapy in combination with immune checkpoint inhibitors in this *TFE3*-tRCC subtype.

In contrast to *ASPSCR1-TFE3* fusion tRCC, other relatively common *TFE3*-tRCC fusion subtypes were characterized by decreased angiogenesis gene signatures. These findings provide a molecular explanation for the unfavorable clinical outcome to antiangiogenic monotherapy in ours and previous clinical studies[27,28]. Tumors in cluster 1 and 3 were associated with moderate stroma signatures. Tumors in cluster 2 and 4 showed an enrichment of KRAS down signature and E2F pathway, respectively. In addition, both ours and previous studies showed the activation of autophagy, mTOR and proliferation signaling in the whole *TFE3*-tRCC[28–30]. Therefore, targeting these specific aberrations, such as stromal disruptors, E2F, autophagy, mTOR, and proliferation inhibitors may be options for patients with advanced *TFE3*-tRCCs.

The current study is not devoid of limitations. Due to the low incidence of *TFE3*-tRCC and its high heterogeneity, it is difficult to collect an adequate number of cases with different subtypes for further analysis. However, our study represents the largest cohort to date investigating the molecular features of *TFE3*-tRCC. Additionally, this was a single-center study thus selection bias is unavoidable. Although with a relatively long follow-up time, we acknowledge that the low proportion of death cases may restrict the statistical power of survival analysis in our study. Future multicenter studies with larger sizes are needed to shed further light on the molecular mechanisms of *TFE3*-tRCC.

Overall, the current study expanded our knowledge of the genomic and transcriptional landscape of *TFE3*-tRCC, emphasized the importance of fusion partners and structures for patient outcomes, and provided a molecular basis for the heterogeneous clinical phenotype. Our study represented a step forward in understanding the *TFE3*-tRCC biology. We expect that our findings will provide a genetic basis for developing personalized therapies for this disease.

## Methods

**Patient identification**. A total of 4581 cases diagnosed as RCC who underwent surgery for the kidney at our center between 2009 and 2019 were reviewed by two experienced uropathologists (Ni Chen and Mengni Zhang). Among them, 1006 suspicious non-clear cell RCC cases were identified via morphological evaluation and were selected for further TFE3 IHC. As a result, 68 TFE3 positive cases were confirmed as *TFE3*-tRCC by break-apart fluorescence in situ hybridization (FISH) assay (Supplementary Figure 1). Untreated primary tumor tissues and adjacent normal samples were collected. RNA-seq and WES of formalin-fixed paraffin-embedded (FFPE) tissues were subsequently performed. RNA-seq was performed on 63 tumors and adjacent normal tissues. WES was performed on 53 FFPE tumor tissues and matched adjacent normal ($n = 42$)/blood ($n = 11$) samples. The study was conducted in accordance with the Declaration of Helsinki and was approved by the Ethics Committee of West China Hospital of Sichuan University. All patients or family members provided written consent for genetic analysis.

**Clinicopathological characteristics and outcomes**. Clinicopathological data were retrospectively collected, including age, gender, tumor size, TNM stage, morphological features, ISUP grade, and systemic treatment strategies. For patients with localized disease, regular evaluations were performed every 6–12 months post-surgery. For those with metastatic disease, regular evaluations were carried out every 4–6 weeks after receiving systematic therapy. For patients receiving systemic treatments, PFS was defined as the time from treatment initiation to disease progression or death. Tumor response was defined by Response Evaluation Criteria in Solid Tumors (RECIST) version 1.1[31]. OS was defined as the time from surgery to the date of death.

**IHC and multiple immunofluorescence**. IHC and multiple immunofluorescence were performed as previously described[32]. Commercially available primary anti-TFE3 (clone MRQ-37, 1:100, MXB biotechnologies, Fujian, China), anti-HIF1A (1:5000, Novus Biologicals, Colorado, USA), anti-CD31 (clone UMAB30, ready to use, ZSGB-BIO, Beijing, ChinaMXB), CD8 (clone C8/144B, ready to use, Dako, Copenhagen, DEN) and PD-L1 (clone 22C3, 1:50, Dako) were used in this study. Multiplex immunofluorescence staining was performed with a PANO 7-plex kit (0004100100, Panovue, Beijing, CHN). CD8, macrophage, and NK cells expression were quantified as positive cell density (cell number per mm²). PD-L1 expression was assessed by tumor proportion score, which was defined as the percentage of tumor cells with membranous PD-L1 staining. PD-L1 expression >1% was defined as positivity.

**Fluorescence in situ hybridization assay**. FFPE tissue sections were examined by using interphase FISH to investigate the rearrangement of the *TFE3* region with an LSI dual-color break-apart probe (GSP *TFE3*, Anbiping company, Guangzhou, China). *TFE3* gene rearrangement would result in break-apart of the normal fused green-red signals, resulting in one green/one red break-apart signal pattern (male), or one green/one red break-apart signal and one remaining normal fused green-red signal pattern (female) in a normal cell. One hundred nonoverlapping tumor nuclei were evaluated. The cut-off value was 10%[33].

**Fusion validation by RT-PCR**. cDNA was synthesized using PrimeScript™ RT Master Mix (RR036A, Takara, Shiga, Japan) according to the manufacture's instruction and then subjected to PCR reactions. Primer sequences for fusion validations are listed in Supplementary Table 5. PCR was performed with the

following thermal cycling conditions: a 98 °C for 10 s; 35 cycles of 95 °C for 10 s, 52 °C for 30 s and 72 °C for 1 min; and a 72 °C for 5 min. PCR products were separated by electrophoresis in agarose gels, purified with ChargeSwitch™ PCR Clean-Up Kit (CS12000, Invitrogen, Oberhausen, GER) and then sequenced by ABI 3730XL automatic sequencer (Life Technologies).

**Total RNA isolation, library preparation, and sequencing**. Total RNA was isolated from each sample (63 tumor samples and 14 paired adjacent normal samples) using the Qiagen RNeasy formalin-fixed paraffin-embedded (FFPE) Kit (73504, Qiagen, Hilden, Germany), following the protocol from the manufacturer. The purity and quantity of total RNA were measured by Nanodrop. The integrity of RNA was evaluated using the RNA Nano6000 Assay Kit on the Bioanalyzer 2100 system (Agilent Technologies, CA, USA). 1 µg RNA of per sample was used as input for the RNA sample preparations. Strand-specific RNA sequencing libraries were generated using the Whole RNA-seq Lib Prep kit for Illumina (RK20303, ABClonal, Shanghai, China). Library quality was evaluated on the Agilent Bioanalyzer 2100 system (Agilent, USA). Final libraries were sequenced at the Novogene Bioinformatics Institute (Beijing, China) on an Illumina Hiseq X10 platform by 150 bp paired-end reads.

**Gene expression quantification and fusion detection**. The raw RNA-sequencing reads were filtered by FastQC, Reads were aligned using STAR (v2.7.0 f)[34] with default parameters to the Ensembl human genome assembly GRCh37. Gene expression levels were estimated by raw count and Transcripts Per Kilobase Million (TPM). Annotations of mRNA in the human genome were retrieved from the GENCODE (v19) database. Paired trimmed/clipped and de-duplicated RNAseq reads were used to identify gene fusion events, and the aligned output was used as input to STAR-Fusion (v1.9.1) using the developer-supplied gencode v33 CTAT library from April 6, 2020. Fusion gene supported by at least two reads were selected.

**Analysis of differentially expressed genes (DEG) and enrichment of signaling pathways**. DEGs were determined using the R package "limma" with cut-off P value < 0.05[35]. Upregulated genes and downregulated genes were used to perform ontology and pathway enrichment analysis based on Gene Ontology and KEGG databases using R package "ClusterProfiler"[36].

**Gene set enrichment analysis**. DEG analysis results were used in GSEA analyses. For gene set analysis, hallmark gene sets from Msigdb database[37] were collected. The top 1500 DEGs were chosen by P value ranked by log2 fold change, and then fed into "fgsea" R packages[38].

**Non-negative matrix factorization clustering**. We selected 1500 genes with the highest variability across tumors, using Median Absolute Deviation (MAD) analysis. Subclasses were then computed after reducing the dimensionality of the expression data from thousands of genes to a few metagenes using consensus NMF clustering. This method computes multiple k-factor factorization decompositions of the expression matrix, and evaluates the stability of the clustering result using the cophenetic coefficient index. The robust consensus NMF clustering of 54 patient tumors using the top 1500 variable genes, and $k = 5$ was identified through testing $k = 2$ to $k = 10$.

**Gene signatures**. Marker genes for renal cell carcinoma immune cell types were obtained from Wang et al[18]. The T-cell infiltration score (TIS) was defined as the mean of the standardized values for nine T-cell subtypes, and the immune infiltration score (IIS) for a sample was similarly defined as the mean of the standardized values for innate and adaptive immune scores[39]. A previously published angiogenesis signature was utilized to measure an angiogenesis score[40]. And epithelial-mesenchymal transition (EMT) score was calculated following the method mentioned in Gibbons et al.[41].

**Implementation of single-sample gene set enrichment analysis (ssGSEA)**. ssGSEA was used for quantifying immune infiltration and activity in tumors using markers reported by Wang et al[18]. The ssGSEA method is an extension of the GSEA[42] method, which works at a single-sample level rather than a sample population. Normalized RNA-Seq data were used as input without further processing (i.e., no standardization or log transformation).

**Quantitative set analysis for gene expression (QuSAGE)**. To reveal latent biological pathways of NMF clustering, we conducted QuSAGE (R qusage v2.20.0) to compare each cluster to all others, leveraging MSigDb hallmark gene sets to identify enriched pathways within each cluster.

**DNA extraction**. Representative sections of formalin-fixed paraffin-embedded (FFPE) tumor ($n = 53$) and matched normal tissues ($n = 42$) or blood samples ($n = 11$) were collected. Tumor sections were reviewed by a pathologist to ensure tumor sections with >70% tumors and <10% necrosis. High-quality genomic DNA

was extracted by using the GeneRead DNA FFPE Kit (180134, QIAGEN, Hilden, GER) according to the manufacturer's instructions. Germline DNA (gDNA) was extracted from white blood cells using the Blood Genomic DNA Mini Kit (CW2087, Cwbiotech, Beijing, China).

**Whole-exome sequencing**. WES was used to determine the mutational landscape of *TFE3*-tRCC. Exome capture was performed using the Agilent SureSelect Human All ExonV5 kit (Technologies, CA, USA) according to the manufacturer's instructions. This was followed by paired-end sequencing using Illumina Novaseq6000 sequencer (Illumine Inc, CA, USA). Mean clean sequence data obtained was 38.30 Gb for tumor samples and 14.95 Gb for normal tissue/blood samples. Details of somatic and germline mutation analysis are described below.

**Read alignment and BAM file generation**. Clean reads were aligned to the reference human genome hg19 (Genome Reference Consortium GRCh37) using the BWA (v0.7.17) (Burrows-Wheeler Aligner) MEM algorithm with default parameters. BAM was coordinate sorted and PCR duplicates were removed with Sambamba (v0.6.8).

**Postalignment optimization**. After the initial alignment of WES data, we followed GATK (v3.8) Best Practice to process all BAMs from the same patient together for a postalignment optimization process called "co-cleaning" which includes GATK IndelRealigner and BaseQualityScoreRecalibration (BQSR). IndelRealigner performed local realignment to further improve mapping quality across all reads at loci close to indels, and BQSR detected and fixed systematic errors made by the sequencer when it estimated the quality score of each base call[43–45].

**Somatic mutations analysis**. The GATK MuTect2 pipeline was run for paired tumor-normal somatic mutation calling with gnomAD database and a panel of normal made from all normal samples to filter common germline mutations and recurrent technical artifacts. The resulting VCFs were filtered by Mutect2 FilterMutectCalls module, variants outside of the capture kit were removed, and FilterByOrientationBias module was used to filter out false-positive calls from OxoG and FFPE. The resulting somatic SNVs and indels were further filtered according to the flowing criteria[46–49]: read depth ≥10 in both tumor and normal samples, mapping quality ≥40 and base quality ≥20, variants allele frequency (VAF) ≥ 5% and supporting reads ≥5 in tumor, VAF in tumor was ≥5 times than that of the matched normal VAF. Variants were annotated with Oncotator v1.9.9.0. To further avoid miscalling germline variants at least 19 read depth in the normal sample in dbSNP sites. Variants were excluded with minor allele frequency (MAF) > 0.01 in public databases, including 1000 G (May 2013), Exome Sequencing Project (ESP6500) and Exome Aggregation Consortium (ExAC v0.3.1). Variants were filtered which annotated as 3'UTR, 5'UTR, 3'Flank, 5'Flank, IGR, Intron, lincRNA, Silent RNA.

**Somatic mutation signature profiling**. The R package MutationalPatterns[50] (v3.0.1) was used to extract the somatic motifs of these samples. In brief, the somatic motifs for each variant were retrieved from the reference sequence and converted into a matrix. NMF was used to estimate the optimal number of mutation signatures extracted from WES samples. Cosine similarity was calculated to measure the similarity between our identified signatures and COSMIC signatures v3.2 [cancer.sanger.ac.uk/cosmic/signatures].

**Somatic copy number alterations analysis**. FACETS (v0.5.14)[51] was used to estimate tumor cellularity and ploidy from paired tumor and normal WES data, and calculated allele-specific somatic copy number alterations. Copy Number (CN) gains were defined as alterations showing total CN > 2 and CN losses were defined as alterations showing total CN < 2. Arm-level events were defined as any gain or loss occurring in an autosome that involved at least 10% of the arm. To identify significantly focal SCNAs, we used GISTIC2 (v2.0.23)[52], which considers both the frequency and amplitude of every SCNA, was employed with the following modified parameters "-smallmem 1 -broad 1 -brlen 0.7 -cap 1.5 -conf 0.99 -ta 0.2 -td 0.25—armpeel 1 -genegistic 1 -savegene 1 -gcm extreme -js 4 -maxseg 2000 -qvt 0.25 -rx 0". CN gains were defined as alterations showing $0.1 < \log2$ CN ratio $< 0.7$ and CN losses were defined as alterations showing $-1.1 < \log2$ CN ratio $< -0.1$. To measure SCNA burden, fraction of copy number-altered genome (FCA) was calculated by dividing the number of bases in segments with mean log2 CN ratio >0.1 or < −0.1 by the number of bases in all segments. SCNA burden high was defined as ≥ 75[th] of percentile of SCNA burden in the relevant cohort. SCNA burden low was defined as < 75[th] of percentile of SCNA burden.

**Tumor suppressor gene screening**. Tumor suppressor genes (TSGs) were obtained from TSGene version 2.0 (https://bioinfo.uth.edu/TSGene/) and IntOGen (https://www.intogen.org) database. Genes mutated in at least two samples were shown in barplot and TSGs were marked with bold font. KEGG enrichment

analysis was done with those genes. Clonality of mutations was determined based on the cancer cell fraction (CCF) estimated by allele-specific copy number analysis.

**Statistics**. All analyses were conducted using R software (v3.6.0) and SPSS (v16.0). All comparisons for continuous variables were performed using the two-sided Mann–Whitney *U* test for two groups. For categorical variables, Pearson's Chi-squared test with continuity correction or Fisher's exact test was used. Survival analyses were conducted using Kaplan–Meier method and the difference was tested using log-rank. Cox proportional hazards regression (forward likelihood ratio model) was used to determine the independent predictor of OS. All clinicopathological parameters and biomarkers at $P < 0.05$ were then further tested on multivariate Cox regression in three patient cohorts (WES + RNAseq cohort, WES cohort and RNAseq cohort). Least absolute shrinkage and selection operator Cox regression were also performed using all variables in the multivariate analyses to identify optimal predictors of OS. A *P* value less than 0.05 was considered statistically significant.

**Reporting Summary**. Further information on research design is available in the Nature Research Reporting Summary linked to this article.

## Data availability

The WES data generated in this study have been deposited in NODE (The National Omics Data Encyclopedia) database (https://www.biosino.org/node/project/detail/OEP002535, accession numbers OEP002535) and the NCBI Sequence Read Archive (SRA) database (https://www.ncbi.nlm.nih.gov/biosample/SAMN20702299, accession numbers SAMN20702299). The gene expression data reported in this paper were deposited in the Gene Expression Omnibus (GEO) database (https://www.ncbi.nlm.nih.gov/geo/query/acc.cgi?acc=GSE167573, accession numbers GSE167573). The transcriptome data of TCGA-KIRC, KIRP and KICH were collected from the following web-links https://portal.gdc.cancer.gov/projects/TCGA-KIRC, https://portal.gdc.cancer.gov/projects/TCGA-KIRP and https://portal.gdc.cancer.gov/projects/TCGA-KICH, respectively. The remaining data are available within the Supplementary Information.

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

## Acknowledgements

The authors thank all patients involved in this study, as well as their families/caregivers. We offer sincere gratitude to Prof. Allen C. Gao from UC Davis Medical Center for the constructive suggestions and careful revision for this manuscript. Thanks to Mrs. Dan Qin from basebiotech Co., Ltd and Ph.D Menghuan Zhang from GloriousMed Clinical Laboratory (Shanghai) Co., Ltd for providing help for bioinformatic analysis and raw data uploading. This work was supported by the Natural Science Foundation of China (NSFC 81972502, 81902577, 81974398,81872107 and 81872108), China Postdoctoral Science Foundation (2020M673239), Research Foundation for the Postdoctoral Program of Sichuan University (2021SCU12014), Post-Doctor Research Project, West China Hospital, Sichuan University (20HXBH026), 1.3.5 project for disciplines of excellence, West China Hospital, Sichuan University (No.0040205301E21), and Science and Technology Support Program of Sichuan Province (2021YFS0119).

## Author contributions

Study concept and design: S.G.X., J.R.C., J.Y.L., X.X.Y. [1], Z.H.L., P.F.S., N.C., H.Z.; Methodology, acquisition, analysis, or interpretation of the data: S.G.X., J.R.C., J.Y.L., X.X.Y. [1], M.N.Z., J.Y., N.H., L.M.Z., X.M.Z., S.Z., X.M.S., X.X.Y.[2], W.B.Z., B.H.L., X.Y.P., L.N., L.Y., Y.T.C., J.G.Z., H.R.Z., J.D.D., Y.L.S., J.Y.L., R.H., J.D.L., Z.P.W., Y.C.N., D.Q., X.L., Q.Z., Z.H.L., P.F.S., N.C., H.Z.; Financial support: S.G.X., R.H., Q.Z., P.F.S., N.C., H.Z.; Drafting of the manuscript: S.G.X., J.R.C., J.Y.L., X.X.Y. [1]; Critical revision of the manuscript for important intellectual content: S.G.X., J.R.C., J.Y.L., X.X.Y. [1], H.J.H, C.M.A., Z.H.L., P.F.S., N.C., H.Z.; Statistical analysis: S.G.X., J.R.C., J.Y.L., X.X.Y. [1], N.H.; Study supervision: Q.W, X.L., Q.Z., H.J.H, Z.H.L., P.F.S., N.C., H.Z. All authors read and approved the final manuscript. X.X.Y. [1] is representative of Xiaoxue Yin. X.X.Y. [2] is representative of Xiaoxia Yang.

## Competing interests

The authors declare no competing interests.
