## [Peer Review File · Nature Communications]

Integrated exome and RNA sequencing of TFE3-translocation renal cell carcinomaReviewers' Comments:

Reviewer #1:

Remarks to the Author:

Summary:

This work builds on recent efforts to characterize the molecular background of a rare renal cell carcinoma subtype. It provides interesting and useful data on different transcriptomic profiles and shows that different fusion patterns are prognostically relevant. The study further corroborates the previously reported significance of copy number alterations as a prognostic factor in tRCC. It also strengthens the results from previous studies regarding the role of SNVs, showing that they are probably not essential for disease progression, which should be considered for future sequencing efforts. While relatively immune ignorant, the transcriptomically determined expression of PD-L1 – although only a minority of cases was positive on IHC – warrants a correlation with clinical response of advanced tRCC to ICB in further studies. Regarding the five transcriptomic clusters and their proposed difference in response to systemic therapy, no strong conclusions can be drawn from the few cases of metastatic tRCCs undergoing treatment in this cohort (n=8).

Abstract:

p.1, line 64: As stated more in detail below, it seems somewhat of a stretch to say that ASPSCR1 fusion partners in this cohort were likely to benefit from TKI/ICB combination therapy, due to the small case numbers. Therefore, this paragraph should be changed.

Introduction:

p.4, line 95-97: WES was performed on only 53 of these cases. Also, if the aim was to identify potentially effective systemic therapies, this cohort would not be suitable.

Results:

p.5: How many patients underwent partial/radical nephrectomy?

p.5 Have the authors investigated whether the retained exons of the fusion partners of TFE3/TFEB contain common functional elements (e.g. a particular protein domain)? This would help in understanding the function of these fusions.

p.5, line 122: The FUBP1-TFE3 fusion structure was previously reported by the group of Marcon et al.

p.7: How were the cutoffs of retained TFE3 exons determined with regard to their categorization into isoforms?

pp.7 and 8: It seems that the spectrum of detected SNVs was quite heterogeneous. As no clear pattern of mutations can be identified, nor could it in previous studies, the prognostic and therapeutic value of SNVs in tRCC is probably low.

pp.8 and 9 Most of the prior work on the association of somatic mutations/copy number aberrations and aggressiveness in TRCC has focused on arm-level alterations (specifically, 9p loss and 17q gain). The authors don't seem to analyze large-scale aberrations (the GISTIC analysis appears more focal, perhaps I'm missing the relevant text). What is the clinical significance of 9p loss, 17q gain, and other arm-level events in their cohort?

pp.8 and 9 Have the authors evaluated whether mutations of tumor suppressor genes are (a) clonal and/or (b) associated with loss of the other allele through loss-of-heterozygosity or a second mutation?

p.11, line 291: It is notable that only a small number of patients received systemic therapy (n=8). The two cases of clinical and partial radiological response to ICB/TKI combination therapy are interesting. However, the small number of cases (n=2) and the fact, that one patient received pembro/axi as a third-line option, does, to our view, not allow a significant conclusion regarding response. Also, the conclusion that highly angiogenic tumors in cluster 5 respond well to TKIs is difficult due to the fact that these patients also received ICB.

Methods:

p.18, line 447: Case TFE3-68 is reported to be stable for 12 months on third-line ICB/TKI

combination, why is it stated in the methods that PFS was only calculated for first-line treatments?

p.20, line 488: RCP=PCR?

p.25: Was the copy number analysis performed allele-specific? Therefore, did it include cases with loss of heterozygosity? If not, comparability of the results with those from previous studies may be limited

p.25, line 639: What was the cutoff to differentiate between low and high copy number variation burden?

Supplementary Data

(a) It is great to see that the authors are providing the mutation calls for each of their samples in Table S4. Could they provide read counts as well (ie the total depth and alt allele counts in the tumor and in the normal tissue)?

(b) Related to above, it would be extremely valuable to provide files detailing the copy number aberrations in these samples. Ideally, this would include seg files, gene-level calls, and arm-level calls.

(c) Are the BAM files available for others to reuse as a resource?

Reviewer #2:

Remarks to the Author:

In this manuscript Sun et al describe a remarkable cohort of 63 untreated primary TFE3-tRCCs that are extensively and thoughtfully characterized including by whole-exome and RNA sequencing. They report five molecular clusters with distinct angiogenesis, stroma, proliferation and KRAS signatures, which showed association with fusion patterns and prognosis. They find that high angiogenesis/ stroma/ proliferation correlates with ASPSCR1-TFE3 fusions, which are also associated with worse outcomes. They speculate that these tumors may benefit from combination of immune checkpoint and anti-angiogenesis inhibitors.

The manuscript is well written and the authors deserve to be commended for not relying on TFE3 IHC, but also uniformly performing FISH to select their cases. They should similarly be commended for confirming three rare gene fusions by RT-PCR and Sanger sequencing.

Major

The main limitation of the study is that despite the large collection of tRCC tumors available (possibly largest to be reported), when broken down by the type of translocation, the number of tumors drops significantly. Authors also assume that despite different structures, translocations involving the same partner should behave similarly. After adjusting for the partner and structure, the largest group is made up of 12 tumors and second largest of 5. Accordingly, it is difficult to draw robust conclusions or adjust p values for multiple comparisons.

The authors should comment on how much of the worse survival associated with ASPSCR1-TFE3 translocations may be due to the presence of type 2 fusions.

Given the multiple potential predictors of outcome (partnering gene, TFE3 structure, CNA burden, particular CNA (1p13 loss), NMF clusters...) a multivariate analysis should be performed.

The interpretation of the results is also limited by what appears to be extensive censoring.

Conclusions about systemic therapy are quite limited as there are only 8 patients. Accordingly, all discussions about the links to therapy response should be regarded as anecdotal and tempered down.

The authors report a defective mismatch repair signature in some tumors. However, it is unclear that mutations were found in MMR genes. The authors report enrichment of this signature with particular

translocations, but in the absence of mechanism, it is hard to rule out that the correlation is spurious. Also, did any of these patients develop metastatic disease and were they treated with checkpoint inhibitors? T

The absence of TFE3 exon 6 in ZC3H4-TFE3 seems unusual. For this and all other translocations, authors should state that the TFE3 open reading frame is preserved in all the fusions.

Minor

It would be good to complement the data in Fig 3D by looking at empirically-derived TME signatures (Wang et al. Can Discov 2018).

I may have missed this, but authors should provide an excel file with all the mutations identified per sample along with quality metrics pertaining the pathogenic nature, etc.

Authors report increased mutation frequencies for TTN, but this is likely to be a passenger.

Given the PD-L1 protein level findings, it may be fitting to forego a discussion of mRNA levels.

Unclear what the source of normal tissue is as some areas refer to normal kidney, whereas others to blood.

We are grateful for the reviewer's thorough evaluation of our work and appreciate the constructive criticisms. Based on reviewers' insightful comments, we have performed additional analyses and made significant improvements to the manuscript, which we describe point-by-point below.

Point-by-point response to Reviewers' comments:

Reviewer #1 (Remarks to the Author):

This work builds on recent efforts to characterize the molecular background of a rare renal cell carcinoma subtype. It provides interesting and useful data on different transcriptomic profiles and shows that different fusion patterns are prognostically relevant. The study further corroborates the previously reported significance of copy number alterations as a prognostic factor in tRCC. It also strengthens the results from previous studies regarding the role of SNVs, showing that they are probably not essential for disease progression, which should be considered for future sequencing efforts. While relatively immune ignorant, the transcriptomically determined expression of PD-L1 – although only a minority of cases was positive on IHC – warrants a correlation with clinical response of advanced tRCC to ICB in further studies.

We thank the reviewer for their kind comments. We appreciate the reviewer's crucial opinions, and we value them highly. We aim to deliver our points on this manuscript by addressing all the comments thoroughly.

- 1. Regarding the five transcriptomic clusters and their proposed difference in response to systemic therapy, no strong conclusions can be drawn from the few cases of metastatic tRCCs undergoing treatment in this cohort (n=8).**

[Response]

We appreciate the professional comment. As the reviewer recommended, we have altered some of our descriptions concerning treatment recommendations for advanced *TFE3*-tRCC in the Discussion section as shown below and indicated in the revised manuscript. Furthermore, we remove Figure 5C (Potential therapeutic targets for patients with different NMF clusters) in our primary manuscript.

[Revised] Figure 6

Figure 6. Responses to systemic treatment and potential therapeutic targets for patients with *TFE3*-tRCC.

(A) Swimmer plot depicts the PFS of individual patients receiving first line TKIs treatments. Vertical line indicates PFS at 3 months.

(B) Baseline imaging in two patients (TFE3-68 and TFE3-65) before initiation of systematic treatment and after they received the combination of Pembrolizumab plus Axitinib treatment.

TKIs = tyrosine kinase inhibitor, PFS = progression-free survival. Arrow indicates tumor lesion.

[Revised] page 17, line 12-14 in Discussion section

Therefore, our data may support clinical investigation of anti-angiogenic therapy in combination with immune checkpoint inhibitors in this *TFE3*-tRCC subtype.

[Revised] page 17, line 23-25 in Discussion section

Therefore, targeting these specific aberrations, such as stromal disruptors, E2F, autophagy, mTOR and proliferation inhibitors may be options for patients with advanced *TFE3*-tRCCs.

[Revised] page 17, line 15-17 in Discussion section

We expect that our findings will provide a genetic basis for developing personalized therapies for this rare disease.

- 2. p.1, line 64: As stated more in detail below, it seems somewhat of a stretch to say that *ASPSCR1* fusion partners in this cohort were likely to benefit from TKI/ICB combination therapy, due to the small case numbers. Therefore, this paragraph should be changed.**

[Response]

Thank you for the suggestion. We revised this sentence in the Abstract.

[Revised] page 3, line 14-16 in Abstract section

In line with the aggressive nature, the high angiogenesis/stroma/proliferation cluster exclusively consisted of tumors with *ASPSCR1-TFE3* fusion.

Introduction:

- 3. p.4, line 95-97: WES was performed on only 53 of these cases. Also, if the aim was to identify potentially effective systemic therapies, this cohort would not be suitable.**

[Response]

Thank you. We revised the final sentence in the Introduction.

[Revised] page 4, line 23-27 in Introduction section

Therefore, we applied whole-exome sequencing (WES) on 53 *TFE3*-tRCCs and RNA sequencing (RNA-seq) on 63 *TFE3*-tRCCs to reveal their genomic and transcriptomic characteristics and discover molecular mechanisms potentially involved in tumor progression.

Results:

4. p.5: How many patients underwent partial/radical nephrectomy?

[Response]

In our cohort, 27 (39.7%) and 41 (60.3%) patients underwent nephron-sparing surgery and radical nephrectomy, respectively. We have described the operative manners in the Results section and added the data to Table 1.

[Revised] page 5, line 13-16 in Results section

For primary kidney tumors, 27 (39.7%) and 41 (60.3%) patients underwent nephron-sparing surgery and radical nephrectomy, respectively. Ten (14.7%) patients died at the end of follow-up (median 43.8 months, 95% CI: 31.5-56.1).

[Revised] Table 1

Table 1. Baseline clinicopathologic characteristics of *TFE3*-tRCC.

Clinicopathologic characteristics	Total (n=68)
Age, median (range)	32.5 (5-70)
Gender, n (%)	
Male	26 (39.7%)
Female	42 (60.3%)
Tumor size, median (cm, range)	4.7 (1.4-19.6)
T stage, n (%)	
≤T2	59 (86.8%)
≥T3	9 (13.2%)
N stage, n (%)	
N0	52 (76.5%)
N1	16 (23.5%)
M stage, n (%)	
M0	61 (89.7%)
M1	7 (10.3%)
ISUP grade, n (%)	
≤ 2	31 (45.6%)
≥ 3	37 (54.4%)
Nephrectomy, n (%)	
Nephron sparing surgery	27 (39.7%)
Radical nephrectomy	41 (60.3%)

ISUP: The International Society of Urological Pathology.

5. p.5 Have the authors investigated whether the retained exons of the fusion partners of TFE3/TFEB contain common functional elements (e.g. a particular protein domain)? This would help in understanding the function of these fusions.

[Response]

We thank the reviewer for the suggestions that help enhance the value of our study. To investigate the functional domains of retained exons of the partners, Pfam and Uniprot were used to annotate the functional domains of *TFE3* and the fusion partner genes (**Table S2**, see details in Appendix). According to results from the gene fusion analysis, the retained functional domains of *TFE3* and the fusion partners were identified and visualized in revised **Figure 1C**. We found 42% (24/57) of fusion partner genes retain all functional domains. Interestingly, fusion partners that play regulatory roles in mRNA processing and mRNA splicing, including *NONO*, *SFPQ* and *RBM10*, retained all RNA recognition motifs (RRM). We summarized these data in the Results section.

[Revised] Figure 1C

Figure 1C. Exons and functional domains of the TFE3 and fusion partner genes detected in our *TFE3*-tRCC cohort.

AD = strong transcription activation domain, bHLH = basic helix–loop–helix domain, LZ = leucine zipper domain, RRM = RNA-recognition motif, SREBF1 = Sterol Regulatory Element Binding Transcription Factor 1, MAD2L2 = mitotic spindle assembly checkpoint protein MAD2B, KH = K homology domain, Znf = zinc-finger domains.

[Revised] page 6, line 24-29 in Results section

Next, we analyzed the functional domains of the retained exons of the fusion partner genes. We found 42% of fusion partner genes retained all functional domains. Interestingly, fusion partners that play regulatory roles in mRNA processing and/or mRNA splicing, including *NONO*, *SFPQ* and *RBM10*, retained all RNA recognition motifs (RRM).

6. p.5, line 122: The FUBP1-TFE3 fusion structure was previously reported by the group of Marcon et al.

[Response]

As the reviewer mentioned, *FUBP1-TFE3* fusion was previously identified in two cases, most recently by the group of Wang et al.¹ and Marcon et al.², respectively. Wang et al. reported a case with *FUBP1* exon 17 fused with *TFE3* exon 2. We did not find the fusion structure of *FUBP1-TFE3* in the text and appendix from Marcon et al. In our cohort, the case with *FUBP1-TFE3* fusion resulted in a chimeric transcript composed of exons 1-15 of *FUBP1* and exons 3-10 of *TFE3*, which showed different fusion structure to that which Wang et al. reported.

[Reference]

1. Wang XT, et al. RNA sequencing of Xp11 translocation-associated cancers reveals novel gene fusions and distinctive clinicopathologic correlations. *Mod Pathol* 31, 1346-1360 (2018).
2. Marcon J, et al. Comprehensive Genomic Analysis of Translocation Renal Cell Carcinoma Reveals Copy-Number Variations as Drivers of Disease Progression. *Clin Cancer Res* 26, 3629-3640 (2020).

7. p.7: How were the cutoffs of retained TFE3 exons determined with regard to their categorization into isoforms?

[Response]

The functional domains of *TFE3* include a transcription activation (AD) domain spanning exons 4 and 5, a basic helix–loop–helix domain (bHLH) within exons 7–9, and a leucine-zipper domain (LZ) within exons 9–10 exons. The bHLH-LZ domains (exons 7-10) mediate dimerization, DNA binding, and a putative nuclear localization signal (NLS). Yin et al. recently reported that TFE3 can interact with CDK4-CDK6 complex and be phosphorylated at Ser246 (exon 4 of *TFE3*), which results in nuclear export of TFE3¹. Therefore, in our primary manuscript, we class patients into three groups according to whether the phosphorylation site of Ser246 or AD domain (exon 5 of *TFE3*) is retained. Type 1 (24.6%, 14/57) included tumors retained exons 2-10, 3-10 and 4-10 of *TFE3*. Type 2 (22.8%, 13/57) included tumors retained exons 5-10 of *TFE3*. Tumors retained exon 6-10 and 7-10 of *TFE3* were classified as Type 3 (52.6%, 30/57).

In our cohort, 40% (12/30) of cases which retained *TFE3* 6-10 exons (type 3 fusion) were *ASPSCR1-TFE3* fusions. According to the reviewer's suggestion, additional analyses were performed to determine whether *ASPSCR1-TFE3* fusion or type 3 fusion is the prognostic factor for poor overall survival (OS). Since the majority of cases (92%, 12/13) with *ASPSCR1-TFE3* fusions exhibited type 3 fusion, survival analysis was firstly performed on non-*ASPSCR1-TFE3* fusion cohorts to evaluate the prognostic value of type 3 fusion on OS. We found that type 3 fusion was not the predictor for poor OS (P = 0.647, **Supplementary Table, sheet 1**). In order to further elucidate the prognostic value of fusion structure, we subsequently reviewed the literature for all published cases of *TFE3*-tRCC and collected information about fusion type, structure and patient survival (External cohort)¹⁻¹⁰. A total of 53 cases which reported *TFE3*-fusion type and structure were involved in further analysis (**Supplementary Table, sheet 2 and 3**). In line with our results, 30% (10/33) cases with type 3 fusion were *ASPSCR1-TFE3* fusions, and majority (91%, 10/11) of *ASPSCR1-TFE3* fusions retained *TFE3* 6-10 exons. Survival analysis showed that type 3 fusion was not associated with unfavorable OS in both all

patient cohorts and non-*ASPSCR1-TFE3* fusion cohorts. These results suggested that type 3 fusion was not associated with poor OS according to the current evidence. Therefore, we removed the results and definition of fusion types in the modified manuscript.

[Revised] Supplementary Table (sheet 1)

Supplementary Table. Univariate survival analysis of *TFE3* fusion structures in WCH and External cohorts (This table was not included in the manuscript)

Cohort	Covariate	Levels	N	2 year OS	P value
WCH cohort	retained TFE3 exons	6-10 exons vs. other	29 vs. 28	87.6% vs. 91.9%	0.086
External cohort	retained TFE3 exons	6-10 exons vs. other	33 vs. 20	88.9% vs. 82.3%	0.591
WCH cohort + External cohort	retained TFE3 exons	6-10 exons vs. other	62 vs. 48	88.1% vs. 88.0%	0.385
WCH non- ASPSCR1 cohort	retained TFE3 exons	6-10 exons vs. other	17 vs. 27	100.0% vs. 91.6%	0.647
External non- ASPSCR1 cohort	retained TFE3 exons	6-10 exons vs. other	23 vs. 19	89.7% vs. 81.2%	0.585
WCH non- ASPSCR1 cohort + External non- ASPSCR1 cohort	retained TFE3 exons	6-10 exons vs. other	40 vs. 46	93.8% vs. 87.4%	0.998

WCH: West China hospital; External cohort: patients collected from reference reported.

[Reference]

1. Yang C, et al. CDK4/6 regulate lysosome biogenesis through TFEB/TFE3. *Journal of Cell Biology* 219,(2020).
2. Argani P, et al. Primary renal neoplasms with the ASPL-TFE3 gene fusion of alveolar soft part sarcoma: a distinctive tumor entity previously included among renal cell carcinomas of children and adolescents. *Am J Pathol* 159, 179-192 (2001).
3. Sukov WR, et al. TFE3 rearrangements in adult renal cell carcinoma: clinical and pathologic features with outcome in a large series of consecutively treated patients. *Am J Surg Pathol* 36, 663-670 (2012).
4. Ellis CL, et al. Clinical heterogeneity of Xp11 translocation renal cell carcinoma: impact of fusion subtype, age, and stage. *Mod Pathol* 27, 875-886 (2014).
5. Classe M, et al. Incidence, clinicopathological features and fusion transcript landscape of translocation renal cell carcinomas. *Histopathology* 70, 1089-1097 (2017).
6. Wang XT, et al. SFPQ/PSF-TFE3 renal cell carcinoma: a clinicopathologic study

emphasizing extended morphology and reviewing the differences between SFPQ-TFE3 RCC and the corresponding mesenchymal neoplasm despite an identical gene fusion. Hum Pathol 63, 190-200 (2017).

7. Xia QY, et al. Xp11 Translocation Renal Cell Carcinomas (RCCs) With RBM10-TFE3 Gene Fusion Demonstrating Melanotic Features and Overlapping Morphology With t(6;11) RCC: Interest and Diagnostic Pitfall in Detecting a Paracentric Inversion of TFE3. Am J Surg Pathol 41, 663-676 (2017).
8. Xia QY, et al. Xp11.2 translocation renal cell carcinoma with NONO-TFE3 gene fusion: morphology, prognosis, and potential pitfall in detecting TFE3 gene rearrangement. Mod Pathol 30, 416-426 (2017).
9. Fukuda H, et al. A novel partner of TFE3 in the Xp11 translocation renal cell carcinoma: clinicopathological analyses and detection of EWSR1-TFE3 fusion. Virchows Arch 474, 389-393 (2019).
10. Kato I, et al. RBM10-TFE3 renal cell carcinoma characterised by paracentric inversion with consistent closely split signals in break-apart fluorescence in-situ hybridisation: study of 10 cases and a literature review. Histopathology 75, 254-265 (2019).
11. Tretiakova MS, Wang W, Wu Y, Tykodi SS, True L, Liu YJ. Gene fusion analysis in renal cell carcinoma by FusionPlex RNA-sequencing and correlations of molecular findings with clinicopathological features. Genes Chromosomes Cancer (2019).
12. Marcon J, et al. Comprehensive Genomic Analysis of Translocation Renal Cell Carcinoma Reveals Copy-Number Variations as Drivers of Disease Progression. Clin Cancer Res 26, 3629-3640 (2020).

8. pp.7 and 8: It seems that the spectrum of detected SNVs was quite heterogeneous. As no clear pattern of mutations can be identified, nor could it in previous studies, the prognostic and therapeutic value of SNVs in tRCC is probably low.

[Response]

We agree with the reviewer's comment. We evaluated the correlation between frequently mutated genes (*DST*, *DNAH8* and *HMHA1*) and TMB on patient

survival. Consistent with the reviewer's speculation, there was no relation between these SNVs and prognosis (**Table S6**). We added these data in the Results section.

[Revised] page 9, line 5-9 in Results section

In line with previous studies, the mutational spectrum of *TFE3*-tRCC was quite heterogeneous. Survival analysis indicated that there was no relation between frequent SNVs and patient survival (**Table S6**).

[Revised] Table S6

Table S6. Univariate survival analysis of frequent mutated genes and TMB

Covariate	Levels	N	Median OS (months)	2 year OS	Log-rank P
DST mutation	Yes vs. No	5 vs. 46	82.2 vs. 104.1	100% vs. 92.2%	0.842
DNAH8 mutation	Yes vs. No	3 vs. 48	-	100% vs. 92.7%	0.661
HMHA1 mutation	Yes vs. No	4 vs. 47	73.8 vs. 103.8	75.0% vs. 94.8%	0.442
TMB*	High vs. Low	14 vs. 37	112.1 vs. 99.8	93.3% vs. 93.2%	0.631

TMB: tumor mutation burden; * TMB \geq 75th percentile of TMB was defined as TMB high.

9. pp.8 and 9 Most of the prior work on the association of somatic mutations/copy number aberrations and aggressiveness in TRCC has focused on arm-level alterations (specifically, 9p loss and 17q gain). The authors don't seem to analyze large-scale aberrations (the GISTIC analysis appears more focal, perhaps I'm missing the relevant text). What is the clinical significance of 9p loss, 17q gain, and other arm-level events in their cohort?

[Response]

We thank the reviewer for raising this important point. In the initial analysis, we used CNVkit and GISTIC to detect potential somatic copy number alterations (SCNA). As the reviewer mentioned, GISTIC analysis focuses on focal aberrations, and the CNVkit is not usually used for allele-specific copy number alteration analysis. In the revised manuscript, we used FACETS¹ v0.5.14 to

perform allele-specific CNAs analysis. Arm-level events were defined as any gain or loss occurring in an autosome that involved at least 10% of the arm. Copy Number (CN) gains were defined as alterations showing total CN >2 and CN losses were defined as alterations showing total CN < 2. In our cohort, we identified 6 (11.3%) cases with 9p loss and 12 (22.6%) cases with 17q gain. Among them, 9p loss was associated with poor overall survival (OS, log-rank $P < 0.001$), but not for 17q gain (log-rank $P = 0.179$). Tumors with 22q loss were correlated with *ASPSCR1-TFE3* fusion (4/4 vs. 2/38, $P = 0.005$), higher ISUP nuclear grade (ISUP ≥ 3 , 7/19 vs. 0/27, $P = 0.004$) and more frequent lymph node metastasis (5/7 vs. 2/39, $P = 0.004$). Cases with 9p loss were associated with increased lymph node metastasis (4/8 vs. 2/39, $P = 0.019$). Moreover, we found that loss of chromosome arms 1p, 2p, 6q, 8p and 22q was associated with poor OS (**Figure S7 and Table S8**). According to another reviewer's suggestion, multivariate analysis was performed. After adjustment for clinicopathologic features, 22q loss were independent predictor for poor OS ($P = 0.004$, **Table S4 and Figure S4**). These results were added in the Results section. We also changed some text discussing about the clinical significance of SCNA in the Abstract and Discussion sections.

[Revised] Figure S7

Fig.S7

Figure S7. Somatic copy number alterations (SCNAs) associated with overall survival.

(A-E) Overall survival by the status of 6 chromosome regions with loss. (F) Overall survival by the status of CNA burden.

[Revised] Figure S4

Figure S4. Identify potential predictors for overall survival using LASSO cox regression.

(A) LASSO coefficient profiles of 16 prognosticators in WES+RNAseq cohort. (C) LASSO coefficient profiles of 14 prognosticators in WES cohort. (B and D) Cross-validation for turning parameter selection via minimum criteria in the LASSO regression model.

[Revised] Figure 2

Figure 2. The mutational landscape of *TFE3*-tRCC.

(A) Clinical features and molecular data for 53 tumors (rows) are displayed as heatmaps. (B) Frequently mutated genes in the *TFE3*-tRCC cohort. (C) Kaplan–Meier curves show the OS between patients with 22q loss. Focal loss and gain determined from GISTIC 2.0 analysis.

[Revised] Table S4

Supplementary Table 4. Multivariate analyses of clinicopathologic and genomic features in predicting OS

	WES+RNA-seq cohort (n=53, variates=17)		WES cohort (n=53, variates=15)		RNA-seq cohort (n=68, variates=9)	
	HR (95%CI)	P Value	HR (95%CI)	P Value	HR (95%CI)	P Value
Tumor size, cm						
≥ Median vs. < Median	-	-	-	-	7.88(1.24-50.22)	0.029
M stage						
1 vs. 0	35.16(2.58-479.06)	0.008	32.79(2.58-416.11)	0.007	93.08(7.44-1164.81)	<0.001
22p loss						
Yes vs. No	30.32(3.00-306.6)	0.004	52.49(5.58-493.64)	0.001	-	-
ASPSCR1-TFE3 fusion						
Yes vs. No	-	-	-	-	17.42(2.00-151.72)	0.010

OS: overall survival; HR: hazard ratio; CI: confidence interval

WES+RNA-seq cohort: Unadjusted.

WES cohort: Adjusted for Gender, Age, Tumor size, ISUP grade, T stage, N stage, M stage, SCNA burden, 1p loss, 2p loss, 6q loss, 8p loss, 9p loss and 22p loss.

RNA-seq cohort: Adjusted for Gender, Age, Tumor size, ISUP grade, T stage, N stage, M stage, ASPSCR1-TFE3 fusion and NMF cluster.

[Revised] Table S8

Table S8. Baseline clinicopathologic characteristics of different copy number alterations.

Characteristics	1p loss N=11	2p loss N=5	6q loss N=8	8p loss N=6	9p loss N=6	22q loss N=7
Age, median (range)	25 (22-70)	39 (22-70)	31 (22-70)	31 (22-70)	31 (22-69)	24 (22-38)
Gender, n (%)						
Male	3 (27.3)	2 (40.0)	4 (50.0)	4 (66.7)	2 (33.3)	2 (28.6)
Female	8 (72.7)	3 (60.0)	4 (50.0)	2 (33.3)	4 (67.7)	5 (71.4)
Tumor size, median (cm, range)	5.6 (2.1-17.5)	6.5 (3.3-6.9)	5.6 (4.0-17.5)	5.6 (4.9-13.4)	6.2 (5.0-17.5)	5.6 (2.0-17.5)
T stage, n (%)						
≤T2	11 (100.0)	5 (100.0)	7 (87.5)	6 (100.0)	5 (83.3)	6 (85.7)
≥T3	0 (0.0)	0 (0.0)	1 (12.5)	0 (0.0)	1 (16.7)	1 (14.3)
N stage, n (%)						
N0	7 (63.6)	3 (60.0)	6 (75.0)	4 (66.7)	2 (33.3)	2 (28.6)
N1	4 (36.4)	2 (40.0)	2 (25.0)	2 (33.3)	4 (67.7)	5 (71.4)
M stage, n (%)						
M0	10 (90.9)	5 (100.0)	6 (75.0)	6 (100.0)	5 (83.3)	6 (85.7)
M1	1 (9.1)	0 (0.0)	2 (25.0)	0 (0.0)	1 (16.7)	1 (14.3)
ISUP grade, n (%)						
≤ 2	5 (45.5)	3 (60.0)	3 (37.5)	2 (33.3)	1 (16.7)	0 (0.0)
≥ 3	6 (54.5)	2 (40.0)	5 (62.5)	4 (66.7)	5 (83.3)	7 (100.0)
ASPSCR1-TFE3 fusion (n=8)						
No	5 (50.0)	3 (60.0)	6 (85.7)	4 (80.0)	4 (80.0)	2 (33.3)
Yes	5 (50.0)	2 (40.0)	1 (14.3)	1 (20.0)	1 (20.0)	4 (66.7)
TMB*, n (%)						
High (≥ 75 th percentile)	1 (9.1)	0 (0.0)	2 (25.0)	2 (33.3)	3 (50.0)	1 (14.3)
Low (< 75 th percentile)	10 (90.9)	5 (100.0)	6 (75.0)	4 (66.7)	3 (50.0)	6 (85.7)
SCNA burden**, n (%)						
High (≥ 75 th percentile)	8 (72.7)	3 (60.0)	5 (62.5)	4 (66.7)	6 (100.0)	6 (85.7)
Low (<75 th percentile)	3 (27.3)	2 (40.0)	3 (37.5)	2 (33.3)	0 (0.0)	1 (14.3)

ISUP: The International Society of Urological Pathology, TMB: tumor mutation burden; SCNA: somatic copy number alteration. * TMB ≥ 75th percentile of TMB was defined as TMB high. ** SCNA burden ≥ 75th percentile was defined as SCNA burden high.

[Revised] page 3, line 7-10 in Abstract section

ASPSCR1-TFE3 fusion and several somatic copy number alterations, including the loss of 22q, were associated with aggressive features and poor outcomes.

[Revised] page 9, line 10-29 in Results section

The most frequently observed individual arm-level events included gain of 17q (12/53, 23%) and 19p (11/53, 21%), and loss of 19p (16/53, 30%), 14q (14/53, 26%) and 1p (11/53, 21%). The most frequent focal events were gain of 19p13.2 (17/53, 32%), 1q44 (15/53, 28%) and 8q24.3 (13/53, 25%), and loss of 19p12 (15/53, 26%), 14q21.2 (13/53, 25%, **Figure 3C and Table S7**). Previous studies reported that certain copy number events (eg. 9p loss and 17q gain) were correlated with patient outcomes, therefore, we evaluated the association of somatic copy number alterations (SCNA) with clinicopathologic features and prognosis in our *TFE3*-tRCC cohort (**Table S8**). We found that tumors with 22q loss was correlated with *ASPSCR1-TFE3* fusion (4/4 vs. 2/38, $P = 0.005$), higher ISUP nuclear grade (ISUP ≥ 3 , 7/19 vs. 0/27, $P = 0.004$) and more frequent lymph node metastasis (5/7 vs. 2/39, $P = 0.004$). Cases with 9p loss were associated with increased lymph node metastasis (4/8 vs. 2/39, $P = 0.019$). Survival analysis indicated that loss of chromosome arms 1p, 2p, 6q, 8p, 9p and 22q were predictors for poor OS (**Figure S7 and Figure 3D**). Moreover, we identified that tumors with higher SCNA burden significantly correlated with worse survival outcomes (median OS: 59.46 months vs. 111.28 months, $P = 0.006$). After adjustment for clinicopathologic features, 22q loss was identified as an independent predictor for poor OS ($P = 0.004$, **Table S4 and Figure S4**).

[Revised] page 14, line 21-page 15, line 3 in Discussion section

In our cohort, gain of 17q and loss of 9p were the found in 23% and 11% of *TFE3*-tRCC, respectively. Loss of chromosome arm 9p, but not gain of arm 17q was correlated with poor survival. In addition, loss of 1p, 2p, 6q, 8p, 22q and increased SCNA burden were also predictors for poor prognosis. More

importantly, we demonstrated that 22q loss was significantly associated with aggressive clinical features and an independent predictor of worse outcomes for patients with *TFE3*-tRCC. Increased loss of chromosome 22q were observed in type 2 PRCC that encodes *NF2*, *CHEK2* and *SMARCB1*³, which may implicate in carcinogenesis and tumor progression.

[Revised] page 26, line 7-12 in Methods section

FACETS (v0.5.14)¹ was used to estimate tumor cellularity and ploidy from paired tumor and normal WES data, and calculate allele-specific somatic copy number alterations. Copy Number (CN) gains were defined as alterations showing total CN >2 and CN losses were defined as alterations showing total CN < 2. Arm-level events were defined as any gain or loss occurring in an autosome that involved at least 10% of the arm.

[Reference]

1. Shen R, Seshan VE. FACETS: allele-specific copy number and clonal heterogeneity analysis tool for high-throughput DNA sequencing. *Nucleic Acids Res* 44, e131 (2016).
2. Malouf GG, et al. Genomic heterogeneity of translocation renal cell carcinoma. *Clin Cancer Res* 19, 4673-4684 (2013).
3. Ricketts CJ, et al. The Cancer Genome Atlas Comprehensive Molecular Characterization of Renal Cell Carcinoma. *Cell Rep* 23, 313-326 e315 (2018).

10.pp.8 and 9 Have the authors evaluated whether mutations of tumor suppressor genes are (a) clonal and/or (b) associated with loss of the other allele through loss-of-heterozygosity or a second mutation?

[Response]

Thank you for your suggestion. In this revised manuscript, we performed analysis of tumor suppressor genes based on results from mutation and loss-of-heterozygosity. Tumor suppressor genes (TSGs) were obtained from TSGene v2.0 (<https://bioinfo.uth.edu/TSGene/>) and IntOGen

(<https://www.intogen.org>) database. As a result, 105 TSGs related mutations were identified, including 94 SNPs, 9 DELs, and 2 INSs. Clonality of mutations was determined based on the cancer cell fraction (CCF) estimated by allele-specific copy number analysis. Among 105 TSGs mutations, 22 were identified as subclonal alterations and 83 were clonal alterations (CCF \geq 0.9). A total of 5 LOH events were found, and all of them were subclonal mutations (*MAX*, *DNAJB1*, *ERCC2*, *RTN4IP1* and *NOTCH1*). No TSGs were found with a second mutation event according to the current filter criteria. In our *TFE3*-tRCC, 6 TSGs (*BTK*, *CDH1*, *FN1*, *NFATC2*, *NOTCH1* and *NRP1*) were found in at least two samples, of which 75% (9/12) were clonal mutations. All the results were summarized in **Table S5** (see details in Appendix). We also added these data to the Results section.

[Revised] page 9, line 3-6 in Results section

In addition, six tumor suppressor genes previously implicated in cancer (*BTK*, *CDH1*, *FN1*, *NFATC2*, *NOTCH1* and *NRP1*) were found in at least two samples (Figure 3B). Of these, 75% (9/12) alterations were clonal (Table S5).

[Revised] page 27 line 5-6 in Methods section

Clonality of mutations was determined based on the cancer cell fraction (CCF) estimated by allele-specific copy number analysis.

11.p.18, line 447: Case TFE3-68 is reported to be stable for 12 months on third-line ICB/TKI combination, why is it stated in the methods that PFS was only calculated for first-line treatments?

[Response]

Thank you for pointing out our mistake. We have corrected this error.

[Revised] page 19, line 24-27 in Methods section

For patients receiving systemic treatments, progression-free survival (PFS)

was defined as the time from treatment initiation to disease progression or death.

12.p.20, line 488: RCP=PCR?

[Response]

Thank you. This error has been corrected.

[Revised] page 21, line 8-11 in Methods section

PCR products were separated by electrophoresis in agarose gels, purified with ChargeSwitch™ PCR Clean-Up Kit (CS12000, Invitrogen, Oberhausen, GER) and then sequenced by ABI 3730XL automatic sequencer (Life Technologies).

13.p.25: Was the copy number analysis performed allele-specific?

Therefore, did it include cases with loss of heterozygosity? If not, comparability of the results with those from previous studies may be limited

[Response]

Thank you for your professional observation. In the initial analysis, we used CNVkit to perform copy number alteration (CNA) analysis. But the CNVkit is not usually used for allele-specific copy number alteration analysis. In this revised manuscript, we use FACETS (v0.5.14)¹ to perform allele-specific CNA and loss of heterozygosity (LOH) analysis. CN-LOH (Copy neutral LOH) was called if the minor copy number of a segment was equal to 0 and total copy number = 2. DUP-LOH was determined if minor copy number of a segment was equal to 0 and total copy number > 2. Hemizygous deletion LOH was called if minor copy number of a segment was equal to 0 and total copy number = 1. In summary, 309 LOH were detected in 42 *TFE3*-tRCC cases, including 6 CN-LOH, 1 DUP-LOH and 302 hemizygous deletion LOH. CN-LOH and DUP-LOH events occurred in 3p,14q, 9p,10q, and 6q. The detailed information of these LOH events were summarized in **Table S5** (see details in Appendix).

[Reference]

1. Shen R, Seshan VE. FACETS: allele-specific copy number and clonal heterogeneity analysis tool for high-throughput DNA sequencing. *Nucleic Acids Res* 44, e131 (2016).
2. **p.25, line 639: What was the cutoff to differentiate between low and high copy number variation burden?**

[Response]

Sorry for the unclear description. In this study, patients with SCNA burden $\geq 75^{\text{th}}$ of percentile of SCNA burden were defined as SCNA burden high, while those with SCNA burden $< 75^{\text{th}}$ of percentile of SCNA burden were defined as SCNA burden low. We have described the definition in the Methods section.

[Revised] page 26, line 25-27 in Methods section

SCNA burden high were defined as $\geq 75^{\text{th}}$ of percentile of SCNA burden in the relevant cohort. SCNA burden low were defined as $< 75^{\text{th}}$ of percentile of SCNA burden.

15. Supplementary Data

(a) It is great to see that the authors are providing the mutation calls for each of their samples in Table S4. Could they provide read counts as well (ie the total depth and alt allele counts in the tumor and in the normal tissue)?

[Response]

These data were added in **Table S5** (see details in Appendix).

(b) Related to above, it would be extremely valuable to provide files detailing the copy number aberrations in these samples. Ideally, this would include seg files, gene-level calls, and arm-level calls.

[Response]

These data were added in **Table S7** (see details in Appendix).

(c) Are the BAM files available for others to reuse as a resource?

[Response]

We agree that the current next-generation sequencing data should be available for all researchers. Sequencing FASTQ data files have been deposited at the NCBI Sequence Read Archive (SRA) hosted by the NIH (SRA accession: PRJNA701236) and the Gene Expression Omnibus (GEO) database (accession numbers GSE167573). The data are currently ready for publication after acquisition of paper approval.

Reviewer #2, (Remarks to the Author):

In this manuscript Sun et al describe a remarkable cohort of 63 untreated primary TFE3-tRCCs that are extensively and thoughtfully characterized including by whole-exome and RNA sequencing. They report five molecular clusters with distinct angiogenesis, stroma, proliferation and KRAS signatures, which showed association with fusion patterns and prognosis. They find that high angiogenesis/ stroma/ proliferation correlates with ASPSCR1-TFE3 fusions, which are also associated with worse outcomes. They speculate that these tumors may benefit from combination of immune checkpoint and anti-angiogenesis inhibitors.

The manuscript is well written and the authors deserve to be commended for not relying on TFE3 IHC, but also uniformly performing FISH to select their cases. They should similarly be commended for confirming three rare gene fusions by RT-PCR and Sanger sequencing.

We are very grateful for the valuable comments and suggestions. We tried our best to answer the questions and suggestions of reviewer, which we address below.

Major

- 1. The main limitation of the study is that despite the large collection of tRCC tumors available (possibly largest to be reported), when broken down by the type of translocation, the number of tumors drops significantly. Authors also assume that despite different structures, translocations involving the same partner should behave similarly. After adjusting for the partner and structure, the largest group is made up of 12 tumors and second largest of 5. Accordingly, it is difficult to draw robust conclusions or adjust p values for multiple comparisons.**

[Response]

Thank you for your professional comment. We agree with the reviewer's comment that the high heterogeneity of *TFE3*-tRCC has the potential to undermine the conclusions drawn by multivariate analysis. According to the reviewer's suggestion, we performed additional analysis to determine whether fusion partner or structure is the prognostic factor for worse overall survival (OS). Our results showed *ASPSCR1-TFE3* fusion but not fusion structure was associated with poor OS (**see details in Question 2 response**). Therefore, we removed the results and definition of fusion types in the modified manuscript.

In order to find independent predictors for OS, multivariable cox regression analysis was performed in three patient cohorts, including the WES+RNAseq cohort, WES cohort and RNAseq cohort. Considering the relatively low number of samples and high number of variables, least absolute shrinkage and selection operator (LASSO) regression analysis were subsequently performed to minimize bias-variance tradeoff. The results of LASSO regression accord with results of multivariable cox regression (**see details in Question 3 response**). Even so, results from multivariate analyses should be treated with reserve until validated in multicenter studies with larger sample size. This limitation was described in the Discussion sections in our primary manuscript (**page 15, line 28-29 and page 16, line 1 in Discussion section**).

2. The authors should comment on how much of the worse survival associated with *ASPSCR1-TFE3* translocations may be due to the presence of type 2 fusions.

[Response]

Thank you for the professional suggestion. In our cohort, 40% (12/30) cases retained *TFE3* 6-10 exons (type 3 fusion) were *ASPSCR1-TFE3* fusions, we presume the reviewer's question is about whether *ASPSCR1-TFE3* or type 3 fusion is the prognostic factor for worse survival. Therefore, additional analyses were performed to determine whether *ASPSCR1-TFE3* fusion or type 3 fusion is the prognostic factor for poor overall survival (OS) according to the reviewer's

suggestion. Since the majority of cases (92%, 12/13) with *ASPSCR1-TFE3* fusions exhibited type 3 fusion, survival analysis was firstly performed on non-*ASPSCR1-TFE3* fusion cohorts to evaluate the prognostic value of type 3 fusion on OS. We found that type 3 fusion was not the predictor for poor OS (P = 0.647, Supplementary Table). In order to further elucidate the prognostic value of fusion structure, we subsequently reviewed the literature for all published cases of *TFE3*-tRCC and collected information about fusion type, structure and patient survival (**Supplementary Table**, see detail in Appendix)¹⁻¹⁰. A total of 53 cases reported *TFE3*-fusion type and structure were involved in further analysis (External cohort). In line with our results, 30% (10/33) of cases with type 3 fusion were *ASPSCR1-TFE3* fusions, and a majority (91%, 10/11) of *ASPSCR1-TFE3* fusions retained *TFE3* 6-10 exons. Survival analysis showed that type 3 fusion was not associated with unfavorable OS neither in all patient cohorts nor in non-*ASPSCR1-TFE3* fusion cohorts. These results suggested that *ASPSCR1-TFE3* fusion but not type 3 fusion was associated with poor OS according to the current evidence. Therefore, we removed the results and definition of fusion types in the modified manuscript.

[Revised] Supplementary Table (sheet 1, This table was not included in the manuscript)

Supplementary Table. Univariate survival analysis of *TFE3* fusion structures in WCH and External cohorts

Cohort	Covariate	Levels	N	5y OS	P value
WCH cohort	retained TFE3 exons	6-10 exons vs. other	29 vs. 28	58.8% vs. 91.9%	0.086
External cohort	retained TFE3 exons	6-10 exons vs. other	33 vs. 20	71.8% vs. 74.8%	0.591
WCH cohort + External cohort	retained TFE3 exons	6-10 exons vs. other	62 vs. 48	65.2% vs. 85.1%	0.385
WCH non- ASPSCR1 cohort	retained TFE3 exons	6-10 exons vs. other	17 vs. 27	71.4% vs. 91.6%	0.647
External non- ASPSCR1 cohort	retained TFE3 exons	6-10 exons vs. other	23 vs. 19	71.7% vs. 73.1%	0.585
WCH non- ASPSCR1 cohort + External non- ASPSCR1 cohort	retained TFE3 exons	6-10 exons vs. other	40 vs. 46	70.8% vs. 84.4%	0.998

WCH: West China hospital; External cohort: patients collected from reference reported.

[Reference]

1. Argani P, et al. Primary renal neoplasms with the ASPL-TFE3 gene fusion of alveolar soft part sarcoma: a distinctive tumor entity previously included among renal cell carcinomas of children and adolescents. *Am J Pathol* 159, 179-192 (2001).
2. Sukov WR, et al. TFE3 rearrangements in adult renal cell carcinoma: clinical and pathologic features with outcome in a large series of consecutively treated patients. *Am J Surg Pathol* 36, 663-670 (2012).
3. Ellis CL, et al. Clinical heterogeneity of Xp11 translocation renal cell carcinoma: impact of fusion subtype, age, and stage. *Mod Pathol* 27, 875-886 (2014).
4. Classe M, et al. Incidence, clinicopathological features and fusion transcript landscape of translocation renal cell carcinomas. *Histopathology* 70, 1089-1097 (2017).
5. Wang XT, et al. SFPQ/PSF-TFE3 renal cell carcinoma: a clinicopathologic study emphasizing extended morphology and reviewing the differences between SFPQ-TFE3 RCC and the corresponding mesenchymal neoplasm despite an identical gene fusion. *Hum Pathol* 63, 190-200 (2017).
6. Xia QY, et al. Xp11 Translocation Renal Cell Carcinomas (RCCs) With RBM10-TFE3 Gene Fusion Demonstrating Melanotic Features and Overlapping Morphology With t(6;11) RCC: Interest and Diagnostic Pitfall in Detecting a Paracentric Inversion of TFE3. *Am J Surg Pathol* 41, 663-676 (2017).
7. Xia QY, et al. Xp11.2 translocation renal cell carcinoma with NONO-TFE3 gene fusion: morphology, prognosis, and potential pitfall in detecting TFE3 gene rearrangement. *Mod Pathol* 30, 416-426 (2017).
8. Fukuda H, et al. A novel partner of TFE3 in the Xp11 translocation renal cell carcinoma: clinicopathological analyses and detection of EWSR1-TFE3 fusion. *Virchows Arch* 474, 389-393 (2019).
9. Kato I, et al. RBM10-TFE3 renal cell carcinoma characterised by paracentric inversion with consistent closely split signals in break-apart fluorescence in-situ hybridisation: study of 10 cases and a literature review. *Histopathology* 75, 254-265 (2019).
10. Tretiakova MS, Wang W, Wu Y, Tykodi SS, True L, Liu YJ. Gene fusion analysis in renal cell carcinoma by FusionPlex RNA-sequencing and correlations of molecular findings

with clinicopathological features. *Genes Chromosomes Cancer* (2019).

11. Marcon J, et al. Comprehensive Genomic Analysis of Translocation Renal Cell Carcinoma Reveals Copy-Number Variations as Drivers of Disease Progression. *Clin Cancer Res* 26, 3629-3640 (2020).

3. Given the multiple potential predictors of outcome (partnering gene, TFE3 structure, CNA burden, particular CNA (1p13 loss), NMF clusters...) a multivariate analysis should be performed.

[Response]

Thank you for the valuable comments. As the reviewer suggested, we performed multivariate Cox regression analysis including all potential predictors of overall survival (OS, **Table S4**). According to another reviewer's suggestion, allele-specific copy-number aberrant analysis was performed to find the arm-level copy-number aberrant. Survival analysis showed that loss of chromosome arms 1p, 2p, 6q, 8p, 9p and 22q was also associated with worse OS, and all these factors were subjected to multivariate Cox regression. Results showed that 22p loss and metastasis were independent predictors of poor OS (WES+RNA seq-cohort). However, a total of 15 cases, especially four deaths, were excluded in multivariate Cox regression (WES+RNA seq-cohort) due to the absence of WES data. Therefore, multivariate Cox regression analysis were further performed in the WES cohort and RNA-seq cohort, respectively (WES-cohort: including all clinicopathological features and CNA predictors; RNA seq-cohort: including all clinicopathological features, fusion types and NMF cluster). We found that 22p loss and metastasis were still independent predictors of poor OS in the WES-cohort, while larger tumor size, *ASPSCR1-TFE3* fusion and metastasis were independent prognosticators in the RNA seq-cohort. Considering the relatively small sample size and correlations between variables, least absolute shrinkage and selection operator (LASSO) regression analysis were subsequently performed to minimize bias-variance tradeoff. The results of LASSO regression were consistent with results of multivariable cox regression

(Figure S4). We added these results to the Results sections.

[Revised] Table S4

Supplementary Table 4. Multivariate analyses of clinicopathologic and genomic features in predicting OS

	WES+RNA-seq cohort (n=53, variates=17)		WES cohort (n=53, variates=15)		RNA-seq cohort (n=68, variates=9)	
	HR (95%CI)	P Value	HR (95%CI)	P Value	HR (95%CI)	P Value
Tumor size, cm						
≥Median vs. <Median	-	-	-	-	7.88(1.24-50.22)	0.029
M stage						
1 vs. 0	35.16(2.58-479.06)	0.008	32.79(2.58-416.11)	0.007	93.08(7.44-1164.81)	<0.001
22p loss						
Yes vs. No	30.32(3.00-306.6)	0.004	52.49(5.58-493.64)	0.001	-	-
ASPSR1-TFE3 fusion						
Yes vs. No	-	-	-	-	17.42(2.00-151.72)	0.010

OS: overall survival; HR: hazard ratio; CI: confidence interval

WES+RNA-seq cohort: Unadjusted.

WES cohort: Adjusted for Gender, Age, Tumor size, ISUP grade, T stage, N stage, M stage, CNV burden, 1p loss, 2p loss, 6q loss, 8p loss, 9p loss, 22p loss and 1p13.2loss.

RNA-seq cohort: Adjusted for Gender, Age, Tumor size, ISUP grade, T stage, N stage, M stage, *ASPSR1-TFE3* fusion and NMF cluster.

[Revised] Figure S4

Figure S4. Identify potential predictors for overall survival using LASSO cox regression.

(A) LASSO coefficient profiles of 16 prognosticators in WES+RNAseq cohort. (C) LASSO coefficient profiles of 14 prognosticators in WES cohort. (E) LASSO coefficient profiles of 9 prognosticators in RNAseq cohort. (B, D and F) Cross-validation for tuning parameter selection via minimum criteria in the LASSO regression model.

[Revised] page 9, line 27-29 in Results section

After adjustment for clinicopathologic features, 22q loss was identified as an independent predictor for poor OS ($P = 0.004$, **Table S4 and Figure S4**).

[Revised] page 27, line 25-page 28, line 1 in Methods section

All clinicopathological parameters and biomarkers at $P < 0.05$ were then further tested on multivariate Cox regression in three patient cohorts (WES+RNAseq cohort, WES cohort and RNAseq cohort). Least absolute shrinkage and selection operator Cox regression were also performed using all variables in the multivariate analyses to identify optimal predictors of OS.

4. The interpretation of the results is also limited by what appears to be extensive censoring.

[Response]

Thank you for the professional comments. We agree with the reviewer's comment that high censoring rate is a limitation for survival analysis. However, this limitation is likely associated primarily with the high heterogeneity and varied prognosis of *TFE3*-tRCC. In the current study, we screened 4,581 RCC samples at our center between 2009 and 2019 and included all *TFE3*-tRCC in our center. We found *TFE3*-tRCC is a highly heterogeneous disease. Some tumors (7.3%, 5/68) are highly malignant, and patients with these tumor died within 24 months. Also, we noticed that nearly one fifth (12/68) of patients have not met the primary endpoint with more than 7 years follow-up. Therefore, although the median follow-up time reached 43.8 months, only 14.7% (10/68) of patients died at the end of the follow-up. Even so, we will continue to conduct

regular follow-up on these patients and update these results in the future. This limitation is addressed in the Discussion sections in our primary manuscript (page 16, line 4-6 in Discussion section).

5. Conclusions about systemic therapy are quite limited as there are only 8 patients. Accordingly, all discussions about the links to therapy response should be regarded as anecdotal and tempered down.

[Response]

We appreciate the professional comment. As the reviewer recommended, we have modified the text in the Discussion section as follows. Also, we removed Figure 5C (Potential therapeutic targets for patients with different NMF clusters) in our primary manuscript.

[Revised] Figure 6

Figure 6. Responses to systemic treatment and potential therapeutic targets for patients with TFE3-tRCC.

(A) Swimmer plot depicts the PFS of individual patients receiving first line TKIs treatments. Vertical line indicates PFS at 3 months.

(B) Baseline imaging in two patients (TFE3-68 and TFE3-65) before initiation of systematic treatment and after they received the combination of Pembrolizumab plus Axitinib treatment.

TKIs = tyrosine kinase inhibitor, PFS = progression-free survival.

[Revised] page 17, line 12-13 in Discussion section

Therefore, our data may support clinical investigation of anti-angiogenic therapy in combination with immune checkpoint inhibitors in this *TFE3*-tRCC subtype.

[Revised] page 17, line 23-25 in Discussion section

Therefore, targeting these specific aberrations, such as stromal disruptors, E2F, autophagy, mTOR and proliferation inhibitors may be options for patients with advanced *TFE3*-tRCCs.

[Revised] page 18, line 15-17 in Discussion section

We expect that our findings will provide a genetic basis for developing personalized therapies for this rare disease.

6. The authors report a defective mismatch repair signature in some tumors. However, it is unclear that mutations were found in MMR genes. The authors report enrichment of this signature with particular translocations, but in the absence of mechanism, it is hard to rule out that the correlation is spurious. Also, did any of these patients develop metastatic disease and were they treated with checkpoint inhibitors?

[Response]

We thank the reviewer for raising this important point. We used MutationalPatterns (v3.0.1)¹ for re-analysis of mutational signatures. Firstly, the count of somatic mutations was calculated for each type of substitution (96 trinucleotide mutation contexts) to generate the mutational matrix. Then we used Non-negative Matrix Factorization (NMF) to estimate the optimal number of mutation signatures extracted from the data. In our cohort the optimal solution contained three signatures (**Figure S6**), which were then compared to COSMIC signatures version 3.2 [cancer.sanger.ac.uk/cosmic/signatures] using

cosine similarity. Signature A is similar to SBS87 and SBS1 (cosine similarity = 0.81 and 0.78 respectively). This signature was also found to be similar to SBS6 (cosine similarity = 0.76), the defective DNA mismatch repair signature. However, point mutation and indel analysis did not identify aberrations in the MMR or polymerase genes in tumors with SBS6. Moreover, IHC for MLH1, MSH2, MSH6 and PMS2 also showed intact protein expression (**Supplementary Figure**). In our cohort, three patients with SBS6 presented with metastasis, and two of them received anti-angiogenic agents but no immunotherapy. Previous study reported that alveolar soft part sarcoma, a type of tumor characterized by *ASPSCR1-TFE3* fusion, shows DNA mismatch repair deficiency signatures (SBS6, SBS15 and SBS26) and sustained partial responses to immune checkpoint inhibitors². However, without adequate genomic and therapeutic evidences, we were not able to determine whether Signature A truly associated with SBS6 in our *TFE3*-tRCC cohort. Therefore, we modified the results and methodology of mutational signature analysis as indicated. We also changed the results of the mutational signature analysis in the Results section.

[Revised] Figure S6

A

B

Figure S6. Mutational signatures analysis.

(A) The metrics plot showed the optimal solution contained three signatures. (B) Mutational signature barplots. Signature A correspond to SBS87 and SBS1, Signature B correspond to SBS40, and Signature C correspond to SBS22. These corresponding signatures are defined by COSMIC mutational signatures v3.2 (<https://cancer.sanger.ac.uk/cosmic/signatures>).

[Revised] Supplementary Figure (This figure was not included in the manuscript)

Supplementary Figure. Representative IHC demonstrating MLH1, MSH2, MSH6 and PSM2 expression in two selected samples with SBS6 (TFE3-52 and TFE3-68) in our cohort. Magnification X200. Scale bar = 100 μ m.

[Revised] Figure 2

Figure 2. The mutational landscape of *TFE3*-tRCC.

(A) Clinical features and molecular data for 53 tumors (rows) are displayed as heatmaps.

[Revised] page 8, line 13-22 in Results section

We extracted three prominent mutational signatures using a non-negative matrix factorization (NMF) algorithm. Signature B shows the largest contribution, which is found to be similar to SBS40, a signature correlated with age in multiple types of cancer¹. Signature A is similar to both SBS87 (thiopurine exposure) and SBS1 (age-related 5-methylcytosine deamination). Signature C, which highly corresponds to SBS22 (cosine similarity = 0.91), is characterized by T>A transversions at CT [A/G] and has been associated with aristolochic acid exposure. We observed Signature C in 28.8% of patients in our cohort, indicating a potential role of aristolochic acid exposure in the development of Chinese *TFE3*-tRCC.

[Revised] page 25, line 24-page 26, line 2 in Methods section

The R package MutationalPatterns¹ v3.0.1 was used to extract the somatic motifs of these samples. In brief, the somatic motifs for each variant were

retrieved from the reference sequence and converted into a matrix. Non-negative Matrix Factorization (NMF) was used to estimate the optimal number of mutation signatures extracted from WES samples. Cosine similarity was calculated to measure the similarity between our identified signatures and COSMIC signatures v3.2 [cancer.sanger.ac.uk/cosmic/signatures].

[Reference]

1. Blokzijl F, Janssen R, van Boxtel R, Cuppen E. MutationalPatterns: comprehensive genome-wide analysis of mutational processes. *Genome Med* 10, 33 (2018).
2. Lewin J, et al. Response to Immune Checkpoint Inhibition in Two Patients with Alveolar Soft-Part Sarcoma. *Cancer Immunol Res* 6, 1001-1007 (2018).
3. Alexandrov LB, et al. The repertoire of mutational signatures in human cancer. *Nature* 578, 94-101 (2020).

7. The absence of TFE3 exon 6 in ZC3H4-TFE3 seems unusual. For this and all other translocations, authors should state that the TFE3 open reading frame is preserved in all the fusions.

[Response]

We apologize for the incorrect description leading to misunderstanding of the result. The open reading frame of *TFE3* gene is from exon 1 to exon 10. The functional domains of *TFE3* include a transcription activation (AD) domain spanning exons 4 and 5, a basic helix–loop–helix domain (bHLH) within exons 7–9, and a leucine-zipper domain (LZ) within exons 9–10 exons. The bHLH-LZ domains (exons 7-10) mediate dimerization, DNA binding, and a putative nuclear localization signal (NLS). We used Pfam and Uniprot to annotate the functional domains of the exons of *TFE3* and the fusion partner genes (**Table S3**, see details in Appendix). According to results from gene fusion analysis, the retained functional domains of *TFE3* and the fusion partners were identified and visualized in revised **Figure 1C**. In our study, all fusion isoforms retained exons 7-10 of the *TFE3* gene, which includes the bHLH-LZ domains, but a part (47.4%,

27/57) of fusion isoforms contained the AD domain. We modified the text in the Results section to better clarify.

[Revised] Figure 1C

Figure 1C. Exons and functional domains of the TFE3 and fusion partner genes detected in our TFE3-tRCC cohort.

AD = strong transcription activation domain, bHLH = basic helix–loop–helix domain, LZ = leucine zipper domain, RRM = RNA-recognition motif, SREBF1 = Sterol Regulatory Element Binding Transcription Factor 1, MAD2L2 = mitotic spindle assembly checkpoint

protein MAD2B, KH = K homology domain, Znf = zinc-finger domains.

[Revised] page 6, line 5-13 in Results section

All fusion genes preserved the open reading frame between partner genes and the 3' end of *TFE3*. According to the retained exons and functional domains of the *TFE3*, six types of isoforms were found, including retained fragment of *TFE3* 2-10 exons (5.3%, 3/57), 3-10 exons (3.5%, 2/57), 4-10 exons (15.8%, 9/57), 5-10 exons (22.8%, 13/57), 6-10 exons (50.9%, 29/57) and 7-10 exons (1.7%, 1/57). All fusions retained exons 7-10 of the *TFE3* gene, containing the helix-loop-helix (bHLH) and leucine zipper (LZ) domains, but only a part (47.4%, 27/57) of fusion isoforms contained the transcription activation (AD) domain.

Minor

8. It would be good to complement the data in Fig 3D by looking at empirically-derived TME signatures (Wang et al. Can Discov 2018).

[Response]

Thank you for the valued suggestion. We analyzed the tumor microenvironment using the eTME signatures¹ (**Figure 4D**). In line with previous results, obviously low levels of CD8⁺ T cell, T cell and macrophage signatures were identified in *TFE3*-tRCC relative to KIRC. Different from previous results, we found that compared with TCGA RCC subtypes, the T helper 2 cell (Th2) signature was increased expression in *TFE3*-tRCC, while activated dendritic cell (aDC) and plasmacytoid dendritic cell (pDC) signatures had decreased expression (**Figure S8B**). Moreover, natural killer cell (NK) signature was increased in most *TFE3*-tRCC compared with KIRP and KICH. Therefore, the text has been modified in the Results and Methods sections as follows.

[Revised] Figure 4D

Figure 3D. Unsupervised clustering of samples from the TCGA-RCC and our *TFE3-tRCC* cohorts using ssGSEA scores from 25 immune cell types, IIS and TIS.

[Revised] Figure S8B

Figure S8B. Differential expression for each gene signature was additionally analyzed

between the TCGA-RCC and our *TFE3*-tRCC cohorts.

[Revised] page 11, line 2-9 in Results section

Using a refined RCC immune cell gene-specific signatures, we found that compared with TCGA RCC subtypes, the T helper 2 cell (Th2) signature was increased in *TFE3*-tRCC, while the activated dendritic cell (aDC) and plasmacytoid dendritic cell (pDC) signatures had decreased expression (Figure 4 and Figure S8). Moreover, natural killer (NK) signature was increased in most *TFE3*-tRCC compared with KIRP and KICH. Furthermore, obviously low levels of CD8⁺ T cell, T cell and macrophage signatures were identified in *TFE3*-tRCC relative to KIRC (**Figure 4E**).

[Revised] page 23, line 16-17 in Methods section

ssGSEA was used for quantifying immune infiltration and activity in tumors. Marker genes for renal cell carcinoma immune cell types were obtained from Wang et al¹.

[Reference]

1. Wang T, et al. An Empirical Approach Leveraging Tumorgrafts to Dissect the Tumor Microenvironment in Renal Cell Carcinoma Identifies Missing Link to Prognostic Inflammatory Factors. *Cancer Discov* 8, 1142-1155 (2018).

9. I may have missed this, but authors should provide an excel file with all the mutations identified per sample along with quality metrics pertaining the pathogenic nature, etc.

[Response]

Thank you for the suggestion. We have added these data in **Table S5** (see details in Appendix).

10. Authors report increased mutation frequencies for *TTN*, but this is likely to be a passenger.

[Response]

We deleted *TTN* in the list of frequently mutated genes in revised **Figure 3B**.

[Revised] **Figure 3B**

Figure 3B. Frequently mutated genes in the *TFE3*-tRCC cohort. The red dashed line denotes three mutated patients. Tumor suppressor genes are labeled with bold font.

[Revised] page 8, line 29-page 9, line 3 in Results section

The frequently mutated genes (frequency of more than four samples) included *DST*, *DNAH8* and *HMHA1*, whereas the mutated loci at each gene were not recurrent (**Figure 3B** and **Table S5**).

11. Given the PD-L1 protein level findings, it may be fitting to forego a discussion of mRNA levels.

[Response]

We agree with the reviewer's comments and have deleted these contents from the Discussion section.

[Revised] page 15, line 10-18 in Discussion section

Our results demonstrated an immune ignorant TIME in the majority of *TFE3*-tRCCs, characterized by low PD-L1 expression and low CD8+ T cell infiltration in tumor stroma. Except for tumors with *MED15-TFE3* fusion, a low PD-L1 positivity rate was detected in most of our *TFE3*-tRCCs.

12. Unclear what the source of normal tissue is as some areas refer to normal kidney, whereas others to blood.

[Response]

We apologize for the unclear description. WES was performed on 42 FFPE tumor and matched adjacent normal tissues. For 11 tumor samples unavailable to get matched adjacent normal tissues, blood samples were collected for WES. We added a description about the samples using as germline control in Methods sections for clearer description.

[Revised] page 19, line 12-13 in Methods section

WES was performed on 53 FFPE tumor tissues and matched adjacent normal (n=42)/blood (n=11) samples.

We thank all the reviewers and the editors in advance for helping us to improve our manuscript. With the inclusion of these changes, we hope that this manuscript is now suitable for publication in ***Nature Communication***.

Your Sincerely,

Hao zeng

Reviewers' Comments:

Reviewer #1:

Remarks to the Author:

I commend the authors on a thorough and excellent response to my comments and concerns. I appreciate the effort that they put into addressing each point, and I have no further concerns. I also thank them for depositing their sequencing data in public repositories, this will both increase the impact of their paper and serve as a broader resource for others studying this disease.

In the spirit of transparency, I sign my name below.
Ed Reznik

Reviewer #2:

Remarks to the Author:

My queries have been largely addressed and I support moving towards publication.

A few minor points for the authors to improve readability:

51 - probably better "We describe..."

112 - unclear whether adjacent normal was used (Fig says differently)

124/125 - given rarity of cases, I would tone down

128 - do you mean 5'?

134 - "but only a... subset..."

215 - compared to what?

359 - change "highest malignancy" to something else

Point-by-point response to Reviewers' comments:

Reviewer #2 (Remarks to the Author):

My queries have been largely addressed and I support moving towards publication.

We appreciate your comments. Thank you!

A few minor points for the authors to improve readability:

1. 51 - probably better "We describe..."

[Response]

Thank you. We have revised the tense as suggested. Also, we use the present tense to discuss the current work in the abstract section according to the formatting instructions.

[Revised] page 3, line 4 in Abstract section

We describe comprehensive molecular characteristics of 63 untreated primary *TFE3*-tRCCs based on whole-exome and RNA sequencing.

2. 112 - unclear whether adjacent normal was used (Fig says differently)

[Response]

Sorry for the unclear description. RNA-seq was performed on 63 tumors and 14 adjacent normal kidney tissues. We have added this information in the revised manuscript and **Supplementary Figure 1**.

[Revised] page 5, line 20-21 in Results section

RNA-seq was performed on 63 *TFE3*-tRCC tumors and 14 adjacent normal kidney tissues.

[Revised] Supplementary figure 1

3. 124/125 - given rarity of cases, I would tone down

[Response]

We have changed this sentence.

[Revised] page 5, line 20-21 in Results section

Patients with *SETD1B-TFE3* and *ZC3H4-TFE3* fusion developed metastasis by the end of follow-up.

4. 128 - do you mean 5'?

[Response]

Sorry for the unclear description. We have modified this sentence.

[Revised] page 6, line 6-7 in Results section

All fusion genes preserved the open reading frame between the 5' terminal of partner genes and the 3' terminal of *TFE3*.

5. 134 - "but only a... subset..."

[Response]

We have revised this sentence according to your suggestion.

[Revised] page 6, line 11-14 in Results section

All fusions retained exons 7-10 of the *TFE3* gene, containing the helix-loop-helix (bHLH) and leucine zipper (LZ) domains, but only a subset (47.4%, 27/57) of fusion isoforms contained the transcription activation (AD) domain.

6. 215 - compared to what?

[Response]

Sorry for the unclear description. We have added this information in the revised manuscript.

[Revised] page 9, line 10-12 in Results section

Analysis of differentially expressed genes (DEG) identified a total of 3,124 over-expressed and 2,143 under-expressed genes in *TFE3*-tRCCs compared to adjacent normal tissues (Figure 4A and Supplementary Data 5).

7. 359 - change "highest malignancy" to something else

[Response]

We have deleted "highest malignancy" and directly stated tumors with *ASPSCR1-TFE3* fusion had worse survival.

[Revised] page 14, line 21-23 in Discussion section

We observed that tumors with *ASPSCR1-TFE3* fusion had much worse survival compared to those with other fusions.

We thank all the reviewers and the editors in advance for helping us to improve our manuscript. With the inclusion of these changes, we hope that this manuscript is now suitable for publication in ***Nature Communication***.

Your Sincerely,

Hao zeng